# Sub-surface processes and heat fluxes at coarse blocky Murtèl rock glacier (Engadine, eastern Swiss Alps): Seasonal ice and convective cooling render rock glaciers climate-robust

Dominik Amschwand[1,4], Jonas Wicky[1], Martin Scherler[1,†], Martin Hoelzle[1], Bernhard Krummenacher[2], Anna Haberkorn[2], Christian Kienholz[2], and Hansueli Gubler[3]

[1]Department of Geosciences, University of Fribourg, Fribourg, Switzerland
[2]GEOTEST AG, Zollikofen/Bern, Switzerland
[3]Alpug GmbH, Davos, Switzerland
[4]now at: Department of Computer Sciences, University of Innsbruck, Innsbruck, Austria
[†]deceased, 4 June 2022

**Correspondence:** Dominik Amschwand (dominik.amschwand@uibk.ac.at)

**Abstract.** We measured sub-surface heat fluxes and calculated the energy budget of the coarse blocky active layer (AL) of Murtèl rock glacier, a seasonally snow-covered permafrost landform located in the eastern Swiss Alps. In the highly permeable AL, conductive/diffusive heat transfer including thermal radiation, non-conductive heat transfer by air circulation (convection), and heat storage changes from seasonal build-up and melting of ground ice shape the ground thermal regime. Individual heat fluxes are quantified based on a novel in-situ sensor array in the AL (operational in 2020–2023) and direct observations of the ground ice melt (in thaw seasons 2022–2024). The AL energy budget yields the first field-data based quantitative estimate of the climate sensitivity of rock glaciers. The Murtèl total AL heat uptake during the thaw season has been increasing by $4-10\,\mathrm{MJ\,m^{-2}}$ per decade ($4-11\,\%$ of the 2022 heat uptake of $94\,\mathrm{MJ\,m^{-2}}$), driven by earlier snow melt-out in June and increasingly hot–dry July–September periods. Two thaw-season processes render Murtèl rock glacier comparatively climate-robust. First, the AL intercepts $\sim 70\%$ ($55-85\,\mathrm{MJ\,m^{-2}}$) of the thaw-season ground heat flux by melting ground ice that runs off as meltwater, $\sim 20\%$ ($10-20\,\mathrm{MJ\,m^{-2}}$) is spent on heating the blocks, and only $\sim 10\%$ ($7-13\,\mathrm{MJ\,m^{-2}}$) is transferred into the permafrost body beneath and causes slow permafrost degradation. Second, the effective thermal conductivity in the ventilated AL increases from $1.2\,\mathrm{W\,m^{-1}\,K^{-1}}$ under strongly stable temperature gradients (weak warming) to episodically over $10\,\mathrm{W\,m^{-1}\,K^{-1}}$ under unstable temperature gradients (strong cooling), favouring convective cooling by buoyancy-driven Rayleigh ventilation (thermal semiconductor effect). In winter, radiatively cooled air infiltrating through a discontinuous, semi-closed snow cover leads to strong AL cooling. The two characteristic parameters (effective thermal conductivity and intrinsic permeability) are sensitive to debris texture, hence the two undercooling processes are specific to highly permeable coarse blocky material.

# 1 Introduction

The deglaciating high mountains under climate change are now entering a transient phase characterized by strong disequilibria between cryospheric landscape components of differing climate sensitivity and adjustment timescales (Haeberli et al., 2017). Declining seasonal snowpacks (Gottlieb and Mankin, 2024) and vanishing glaciers (Hugonnet et al., 2021) react sensitively and rapidly to the ongoing climatic changes, while shifting vegetation patterns (Körner and Hiltbrunner, 2024) and permafrost warming and degradation (Biskaborn et al., 2019) lag behind. Emerging hazard chains and an altered high-mountain water cycle put ecosystems and communities in mountain areas and downstream lowlands under pressure to adapt (and perhaps to mitigate) (Hock et al., 2022; Hayashi, 2020). In the mountain cryosphere, ice-rich permafrost landforms overlain by a thick, coarse blocky debris layer stand out as the least sensitive, i.e. most robust landforms, appearing as rock glaciers (Haeberli et al., 2006) and frozen talus slopes (Delaloye and Lambiel, 2005) that have been responding slowly to climate change. Note that we use the term "robust" in the sense of "climate-insensitive" (Schaffer and MacDonell, 2022) or "resistant to changes", which is one aspect of resilience (Walker et al., 2004; Jorgenson et al., 2010).

Coarse blocky landforms benefit from specific processes that occur in the clast-supported coarse debris (dm-sized blocks, sparse fine material), collectively known as *undercooling* processes (Wakonigg, 1996; Rist et al., 2003). Undercooling locally creates a stable ground thermal regime (microclimate) typically $1-5\,°C$ colder than the surrounding fine-grained or bedrock terrain (Gorbunov et al., 2004; Harris et al., 1998) and can preserve permafrost conditions at otherwise unfavourable topo-climatic conditions (azonal permafrost) (Morard et al., 2010; Wicky and Hauck, 2020; Wicky et al., 2024). The effect of undercooling has been known for a long time from field investigations (Bächler, 1930; Wakonigg, 1996; Harris and Pedersen, 1998; Kneisel et al., 2000; Gorbunov et al., 2004; Delaloye and Lambiel, 2005; Sawada et al., 2003; Delaloye and Lambiel, 2005; Herz, 2006; Millar et al., 2014; Popescu et al., 2017b; Wagner et al., 2019), and arises from an interplay of several heat transfer and storage processes in a permeable buffer layer between ground and atmosphere or seasonal snow cover (Johansen, 1975; Wakonigg, 1996). Exact processes have long remained elusive. Undercooling is specific to coarse blocky landforms that exhibit a high permeability, pointing at the key role of non-conductive heat transfer by airflow and ice build-up. Climate change has brought these 'cold spots', once known as local curiosities (Balch, 1900; Bächler, 1930; Trüssel, 2013), to the attention of scientists (e.g., hydrologists (Schaffer et al., 2019; Schaffer and MacDonell, 2022; Navarro et al., 2023) and ecologists (Růžička et al., 2012)) and engineers (e.g., artificial passive ground cooling (Guodong, 2005)). By storing frozen and liquid water at seasonal timescales (Jones et al., 2019; Wagner et al., 2021), the hydrological buffer capacity of undercooled permafrost landforms will increasingly contribute to reliable baseflow during droughts, sustaining 'icy seeps' (Tronstad et al., 2016) and wet meadows (Reato et al., 2021) which are refugia for cold-adapted species (Millar et al., 2015; Brighenti et al., 2021). Nonetheless, despite the cold microclimate within their active layer (AL), even these robust undercooled permafrost landforms are not exempt from slow degradation, as exemplarily shown by the Alpine-wide warming (Noetzli and Pellet, 2023), ice loss (Morard et al., 2024), synchronous acceleration (Delaloye et al., 2010; Kellerer-Pirklbauer and Kaufmann, 2012), and in cases destabilisation (Roer et al., 2008; Marcer et al., 2021; Hartl et al., 2023) of rock glaciers. As our mountains

enter uncharted territory, site-specific empirical relations might become invalid. We need quantitative process understanding to anticipate the changes.

On Murtèl rock glacier (Engadine, southeastern Swiss Alps), uncercooling heat transfer processes have been investigated for decades. Large seasonal deviations of the estimated surface energy balance (SEB) were attributed to unmeasured and insufficiently represented non-conductive sub-surface heat transfer processes in the AL (Hoelzle et al., 1999, 2001; Mittaz et al., 2000). First field studies dedicated to the sub-surface (AL) heat transfer processes have been published by Hanson and Hoelzle (2004, 2005) based on their temperature measurements in the uppermost $90\,\mathrm{cm}$ of the AL and previous works (Oswald, 2004; Naguel, 1998). Important insights about the near-surface AL heat transfer processes and interaction with the snow cover have been gained, but the quantitative understanding was insufficient to reliably estimate heat fluxes. Next, the thermal characterization of the coarse blocky material with geophysical methods (electrical resistivity tomography, refraction seismics tomography) has been another important step towards AL heat transfer modelling (Schneider et al., 2012, 2013), that has then been carried out by Scherler et al. (2014). Air circulation and convective heat transfer, long suspected to be the primary heat transfer process to shape the thermal regime in highly permeable coarse blocky debris, have been investigated in the field by (Oswald, 2004; Panz, 2008; Schneider, 2014) and studied numerically by Wicky and Hauck (2017, 2020). In parallel, another important field study on the micro-climate of coarse blocky scree has been carried out by Herz et al. (2003a, b); Herz (2006) in the Matter Valley (western Swiss Alps). They have described the heat transfer processes in detail and estimated heat fluxes and the thermal diffusivity, but for the lack of appropriate measurements could not verify them. One of the few comprehensive data sets beyond ground temperatures in mountain permafrost has been gathered by Rist and Phillips (2005); Rist (2007). They deployed a heat flux plate, ultrasound probes, conductometer, vapour traps and reflectometer probes to characterise the ground hydro-thermal regime of a steep, permafrost-underlain scree slope. To summarize, several heat transfer processes have been successfully simulated separately, for example buoyancy-driven air circulation (Wicky and Hauck, 2017, 2020), purely conductive processes from the interplay between a low-conductive ground and snow cover (Gruber and Hoelzle, 2008), or the interplay between sensible and latent heat storage (Renette et al., 2023). However, few microclimatological studies attempted to simultaneously parametrize all heat fluxes (Mittaz et al., 2000; Hoelzle et al., 2001; Stocker-Mittaz et al., 2002; Hoelzle et al., 2003; Hoelzle and Gruber, 2008; Scherler et al., 2014), and few comprehensive sub-surface hydro-thermal measurements beyond ground temperatures exist in blocky mountain permafrost (Rist et al., 2003; Rist and Phillips, 2005). Also, AL properties like the thermal conductivity are poorly investigated for such coarse blocky material as on Murtèl, where individual blocks have volumes of $\sim 0.1\,\mathrm{m}^3$ (up to $1-10\,\mathrm{m}^3$). Lacking better knowledge, empirical engineering relations developed for sand or gravel were often extrapolated to such large blocks. Without in-situ data, it is unclear whether such extrapolations are valid.

This work, a follow-up study on Amschwand et al. (2024a) where we estimated the SEB, contributes to the quantitative process understanding of heat transfer in the AL of an undercooled, ventilated coarse blocky permafrost landform by presenting an unique data set gained from in-situ measurements on Murtèl rock glacier. Our virtually unparalleled sub-surface measurements (except Rist and Phillips (2005) to our knowledge) in the natural openings between the coarse blocks go beyond common ground temperature recordings and include humidity, airflow speed, thermal radiation, direct heat flux measurements, and stake

measurements of the seasonally falling and rising ground ice table in the coarse blocky AL. We quantitatively describe the two main processes that give rise to the undercooling effect of permeable coarse blocky landforms, namely (i) seasonal build-up and melt of ground ice (seasonal ice turnover), and (ii) convective heat transfer (thermal semiconductor effect (Guodong et al., 2007)). We constrain heat fluxes and heat transfer parameters (effective thermal conductivity) that can be carried into numerical permafrost models. Hence, this work provides insights into the capability and limits of the undercooling effect and on the climate robustness of coarse blocky landforms.

## 2 Study site

The studied Murtèl rock glacier (WGS 84: 46°25′47″N, 9°49′15″E; CH1903+/LV95: 2'783'080, 1'144'820; 2620–2700 m asl.; Fig. 1) is located in a north-facing periglacial area of Piz Corvatsch in the Upper Engadine, a slightly continental, rain-shadowed high valley in the eastern Swiss Alps. Mean annual air temperature (MAAT) is $-1.7°C$, mean annual precipitation is $\sim 900$ mm (Scherler et al., 2014). The tongue-shaped, single-unit (monomorphic sensu Frauenfelder and Kääb (2000)), active rock glacier is surrounded by steep rock faces and is in direct connection with a talus slope (2700–2850 m asl.), $\sim 250$ m long, and $\sim 150$ m wide, and advances slowly onto a permafrost-free, vegetated forefield thinly covered by glacial sediments (till veneer, few large boulders) (Schneider et al., 2013). Murtèl is located at the lower permafrost margin. Crescent-shaped furrows ($\sim 3-5$ m deep) and ridges with steep, in places near-vertical slopes dissect the overall gently northnorthwestward dipping surface ($\sim 10-12°$, $< 15°$, Guodong et al. (2007)) and create a pronounced furrow-and-ridge microtopography in the lowermost part of the rock glacier (Kääb et al., 1998). The snow cover is thicker and lasts longer in furrows than on ridges, influencing the ground thermal regime on a small scale (Bernhard et al., 1998; Keller and Gubler, 1993; Hoelzle et al., 2003). In the colder furrows, the otherwise 3–5 m thick, coarse-grained and clast-supported debris mantle is only $\sim 2$ m thick. The ground ice table is accessible in a few places.

The stratigraphy to a depth of $60$ m is known from several boreholes (Vonder Mühll and Haeberli, 1990; Vonder Mühll, 1996; Arenson et al., 2002, 2010). The coarse blocky AL is 3 m thick on average ($2-5$ m) and consists of large blocks typically with 0.1 to 2 m edge length (Scherler et al., 2014). A few boulders reach dimensions of $\sim 3-5$ m. Fine material ($\leq$sand) is sparse near the surface, but its volume fraction varies laterally and increases with depth (inverse grading, Haeberli et al. (2006)). The AL has a poor water retention capacity. Supra-permafrost water drains within hours–days at the AL base (Tenthorey and Gerber, 1991) and is not considered to significantly modify the ground thermal properties. The vast, connected pores create a high intrinsic permeability (Wicky and Hauck, 2020). Beneath the coarse blocky debris mantle, roughly coinciding with the thermally defined AL, lies the perennially frozen ice-supersaturated rock glacier core. Drill cores have revealed sand- and silt-bearing massive ice (3–28 m depth, ice content over 90% vol.), although boreholes drilled within $\sim 30$ m distance suggest some lateral small-scale heterogeneity (Vonder Mühll and Haeberli, 1990; Arenson et al., 2010).

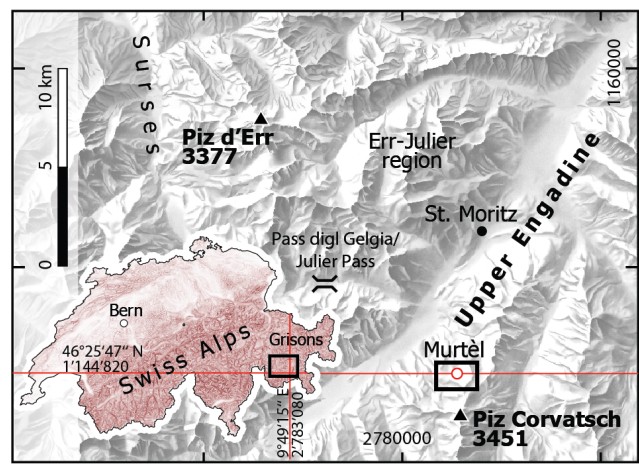

**Figure 1.** Location of Murtèl rock glacier in the Upper Engadine, a high valley in the eastern Swiss Alps (projection: LV95 Swiss coordinate system). Inset map: Location and extent (black rectangle) of regional map within Switzerland (source: Swiss Federal Office of Topography swisstopo). Red northing/latitude line corresponds to both the main/inset map, respectively.

## 3 Measurements and data processing

### 3.1 Field observations and instrumentation

We use measurements from the PERMA-XT sensor cluster presented in Amschwand et al. (2024a), where the above-surface
sensors are described. Most of the below-surface sensors were installed in one natural cavity of the coarse blocky AL that
was large enough for a human to enter and deep enough to come close to the AL base (Figs. 2, 3). This instrumented main
cavity was completely destroyed by rockfall on 20 September 2023. A narrow passage covered by a large block ('lid') lead
into a spacious 'main chamber' with its 'floor' at 2 m depth. A narrow extension reached a depth of 3 m. Its base was covered
by wet fine material (gravel, sand). The cavity in the clast-supported coarse blocky AL was enclosed by large blocks with
voids in between, allowing air circulation. In August 2020, this comparatively large cavity (dimensions $\sim 2 \times 1.5 \times 3$ m) was
instrumented with sensors to measure the temperature, humidity, thermal radiation, heat flux and AL airflow speed at several
depths down to 3 m. Detailed sensor specifications of the sub-surface sensors are given in Table 1, and the locations are shown
in Fig. 3 and Table 2. One thermistor string was suspended in air (TK1), another one (TK6) had its five thermistors drilled 5 cm
into the blocks at depths corresponding to the TK1 thermistors. Relative humidity was measured at two levels, 0.7 (HV5/1)
and 2.0 m (HV5/2) beneath the surface. Three thermo-anemometer recorded wind speed at three levels, close to the surface
(WS/3 at $-0.35$ m), mid-cavity (WS/2 at $-1.5$ m), and in a narrow extension at depth (WS/1 at $-2.1$ m). A back-to-back
pair of pyrgeometer mounted at mid-cavity level (CGR3 at $-1.55$ m) measured the upward and downward thermal radiation
in the cavity. Two heat flux plates were cemented onto the rock surface, one at the underside of a near-surface block (HFP/2
at $-1.1$ m), another one on the cavity 'floor' (HFP/1 at $-2.0$ m). Since accurate distances are required for the calculation of

135 vertical gradients and fluxes, we triangulated the relative height of the sensors in the instrumented cavity with a Leica DISTO X310 laser distance and goniometer.

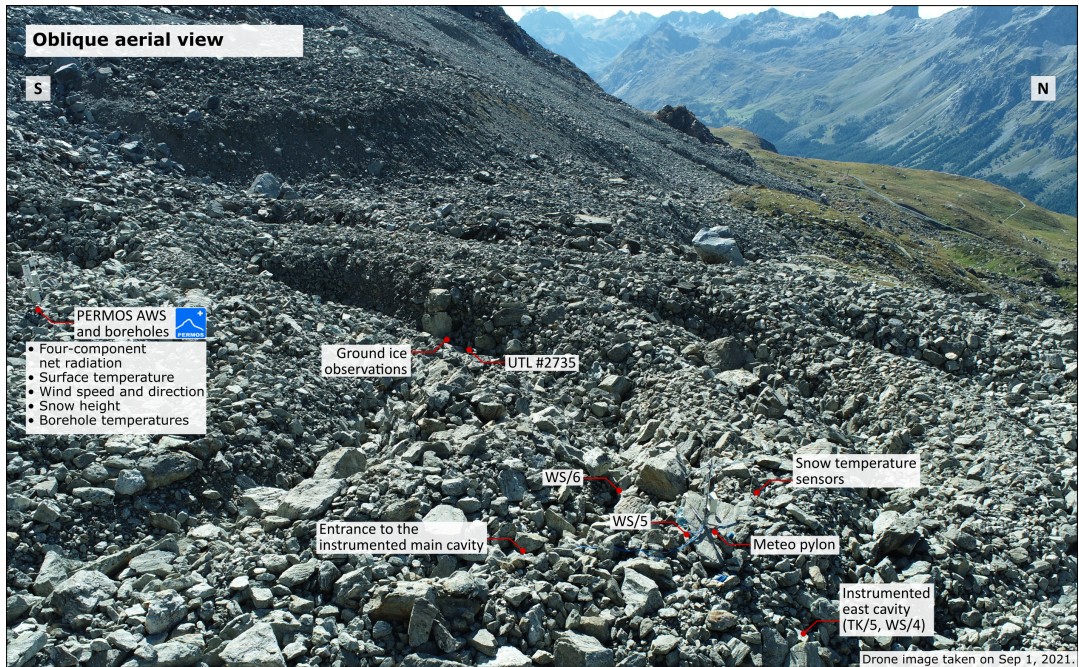

**Figure 2.** Oblique aerial view of the Murtèl rock glacier (foreground) and location of the above-surface sensors and the ground-ice observations in the rock glacier furrow.

Additional thermistor strings and thermo-anemometer were installed in the vicinity at different micro-topographical location to reveal the spatial pattern of temperature and airflow. One additional thermo-anemometer in a similar cavity $20\,\mathrm{m}$ away (WS/4), together with another air-suspended thermistor string (TK5), one near the surface beneath a wind-swept large block
on a ridge (WS/5, 'wind hole'), and one near the surface in a nearby rock glacier furrow (WS/6). The sub-surface sensor array is completed by 10 autonomous miniature temperature loggers (UTL; Table 1) distributed on the rock glacier to grasp the variability of the near-surface ground temperature.

We measured the seasonal ice turnover directly with stake measurements inside the coarse blocky AL. The ground ice is accessible at a few spots at shallow depths of $1-2\,\mathrm{m}$ bgl. (below ground level), all located in furrows where the AL is thinner.
In one spot, a plastic tube was drilled ca. $120\,\mathrm{cm}$ into the ice in August 2009 but subsequently abandoned (C. Hilbich, pers. comm.). We made serendipitous use of it as an 'ablation stake', manually measuring the height of the ground-ice table at each field visit in summer 2022. One nearby autonomous miniature temperature logger (UTL #2735; Fig. 2) measures a near-surface ground temperature $\sim 0.5\,\mathrm{m}$ bgl.

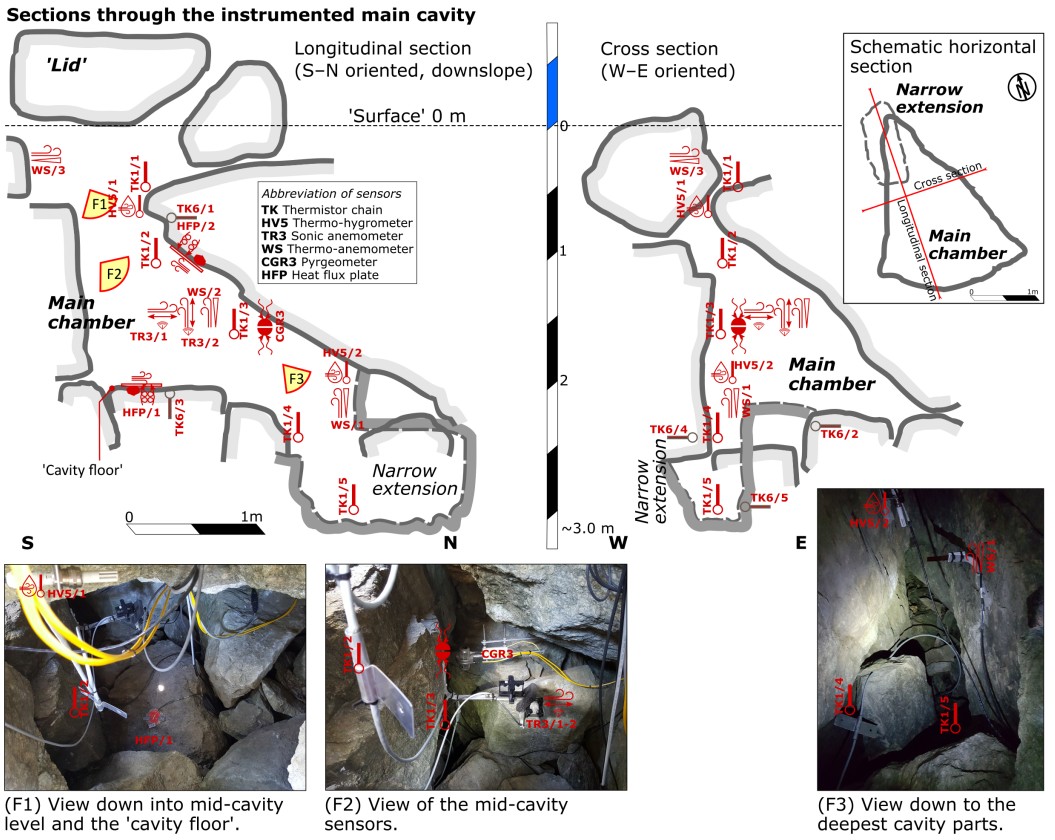

**Figure 3.** Schematic sections and images of the instrumented main cavity with locations of the sensors.

Additionally, we use borehole temperature data provided by the Swiss Permafrost Monitoring Network (PERMOS) for basal conductive heat flux from the rock glacier core and point-wise surface radiation data from the PERMOS automatic weather station (Noetzli et al., 2019).

## 3.2 Data processing

We analyse the data of two years from Sep 1, 2020, to Sep 30, 2022. The above-surface sensors and the corresponding data processing (air temperature, relative humidity, precipitation, snow height) are described in Amschwand et al. (2024a). The used (measurement or meteorological) variables, parameters and constants are tabulated in Appendix Sect. G (Table G1).

**Table 1.** PERMA-XT sub-surface sensor specifications.

| Quantity [unit] | Manufacturer | Sensor type | Accuracy |
|---|---|---|---|
| AL air temperature $T_{al}(z)$ [°C] | TE Connectivity[a] | 44031RC NTC thermistor chain TK1/1–5 | $\pm 0.1$°C |
| AL relative humidity $q_{al}(z)$ [%] | CSI[b] | HygroVUE5 hygrometer | $\pm 0.3$°C; $\leq \pm 4\%$ |
| Rock temperature $T_r(z)$ [°C] | TE Connectivity[a] | 44031RC NTC thermistor chain TK6/1–5 (drilled 5 cm into the blocks) | $\pm 0.1$°C |
| Thermal radiation $Q_{CGR3}^{rad}$ [W m$^{-2}$] | Kipp & Zonen | CGR3 pyrgeometer (4.5−42 μm, FoV 150°) | $< 2$ W m$^{-2}$ |
| Heat flux $Q_{HFP}$ [W m$^{-2}$] | Hukseflux | HFP01 heat flux plate | site-specific |
| Airflow speed proxy[c] [-] | Hukseflux | TP01 thermal properties sensor (formerly WS01) (artificial leaf used as hot-film anemometer) | $\leq \pm 25\%$ or $\leq \pm 0.2$ m s$^{-1}$ |
| Near-surface ground temperature [°C] | GEOTEST | UTL3 miniature temperature logger with YSI 44005 NTC thermistors | $\pm 0.1$°C ($T \leq \pm 20$°C) |

Measurement range and accuracy by manufacturer/vendor. The specifications of the PERMOS sensor are given in Scherler et al. (2014) and Hoelzle et al. (2022).
[a] Thermistor strings manufactured by Waljag GmbH. [b] CSI: Campbell Scientific, Inc.

**Table 2.** PERMA-XT subsurface sensor locations and approximate depth beneath the surface.

| Sensor name | Location | Sensor name | Location |
|---|---|---|---|
| TK1/1–5 | AL air temperature profile TK1/1: −0.5 m; TK1/2: −1.1 m; TK1/3: −1.6 m; TK1/4: −2.4 m; TK1/5: −2.9 m | TK6/1–5 | AL rock temperature profile TK6/1: −0.7 m; TK6/2: −1.8 m; TK6/3: −2.1 m; TK6/4: −2.4 m; TK6/5: −2.9 m |
| HV5/1–2 | AL relative humidity and temperature HV5/1: −0.7 m; HV5/2: −2.0 m | | |
| CGR3 | Thermal radiation in main cavity $Q_{CGR3}^{rad}$ −1.55 m, upward and downward-facing | HFP/1–2 | Heat flux plate in main cavity HFP/2: −1.1 m on underside of near-surface block HFP/1: −2.0 m on 'cavity floor' $Q_{HFP}$ |
| WS/1–6 | Wind-speed proxy WS/3: −0.35 m; WS/2: −1.5 m; WS/1: −2.1 m WS/4: in 'east cavity' −0.3 m WS/5: 'wind hole' at the surface WS/6: in rock glacier furrow −0.3 m | | |
| UTL #2735 | Near-surface air temperature near the 'ablation stake' −0.5 m | | |

The abbreviations correspond to Fig. 3.

### 3.2.1 Sub-surface thermal radiation

The net radiative flux in the AL $Q_{\mathrm{CGR3}}^{rad}$ is calculated from the thermal (long-wave/thermal infrared) radiation of a back-to-back pyrgeometer pair installed in the instrumented cavity (Amschwand et al., 2024a),

$$Q_{\mathrm{CGR3}}^{rad} = (L_{al}^{\downarrow} - L_{al}^{\uparrow}) \tag{1}$$

as the difference between the upwards $L_{al}^{\uparrow}$ and downwards $L_{al}^{\downarrow}$ thermal radiation components. The raw outputs $L_{raw}^{\uparrow/\downarrow}$ of the two pyrgeometers in the instrumented cavity are corrected by accounting for the thermal radiation emitted by the instruments themselves (Kipp & Zonen CGR3 manual, 2014) as in Amschwand et al. (2024a)

$$L_{al}^{\uparrow/\downarrow} = L_{raw}^{\uparrow/\downarrow} + \sigma T_{\mathrm{CGR3}}^4, \tag{2}$$

with the pyrgeometer housing temperature $T_{\mathrm{CGR3}}$. Large ($> 0.5°\mathrm{C}$) or rapid changes of the housing temperature differences
between the back-to-back mounted pyrgeometer hint at dust or water deposition on the upward-facing pyrgeometer window. Such disturbed measurements showed up in the high-resolution (10 minutes) data, but did not significantly affect the daily thermal net radiation balance in the sheltered cavity.

### 3.2.2 Sub-surface airflow speed

We refer to the sub-surface 'wind' in the permeable AL as 'airflow' to differentiate it from the atmospheric wind. Measure-
170 ments of an airflow speed proxy in the AL pore space were performed with six Hukseflux TP01 sensor (formerly WS01; Table 1) consisting of a heated foil that measures a cooling rate, expressed as a heat transfer coefficient $h_{\mathrm{WS}}$ [W m$^{-2}$ K$^{-1}$]. One measurement cycle takes $1\,\mathrm{min}$ and consists of three measurements to detect any offset, one initial measurement before heating, one at $30\,\mathrm{s}$, and a final one after cool-down at $60\,\mathrm{s}$. $h_{\mathrm{WS}}$ is related to airflow speed via $u_{\mathrm{WS}} = (h_{\mathrm{WS}} - 5)/4$. This empirical linearised relation is valid with reasonable accuracy for airflow speeds in the range of $0.1 - 2\,\mathrm{m\,s^{-1}}$ (Hukseflux WS01
manual, 2006). The TP01 sensor does not resolve the direction of the airflow. The deviation of the linearized relation to the common engineering relation $\mathrm{Nu} := hL/k_a = 0.6\,\mathrm{Re}^{0.5}$ (Schuepp, 1993) is within $0.2\,\mathrm{m\,s^{-1}}$ for $u_{\mathrm{WS}} < 0.6\,\mathrm{m\,s^{-1}}$. Hence, the measurements are qualitative rather than absolute. Although the foils were oriented parallel to the dominant airflow direction that can be expected from the local cavity geometry, any airflow perpendicular to the foil creates turbulence that affects the heat transfer efficiency. At very low wind speeds or if the sensor is much warmer than the surrounding air, buoyancy effects become
important relative to forced convection and disturb the airflow speed measurement (increases $h_{\mathrm{WS}}$). Also, deposition and evaporation of liquid water can disturb the measurements (increases $h_{\mathrm{WS}}$), as was revealed by wrapping the heated foils in moist tissues. WS measurements during precipitation events at shallow levels, where water may infiltrate, are discarded. Repeated zero-point checks were performed throughout the snow-free season by enclosing the heated foil in small and dry plastic bags for a few hours, ensuring stagnant conditions with zero airflow speed. Neither a drift nor a temperature dependency beyond the
measurement uncertainty was detected.

### 3.2.3 Heat flux plates

Unlike the CGR3 that measures only thermal radiation, the HFP measures the heat flux across the disk-shaped sensor without discriminating between different heat transfer processes. A limitation is that the small-scale local HFP measurements (sensing area of $8 \times 10^{-4}$ m$^2$) might not be as representative of the average vertical AL fluxes as the CGR3 measurements that hemispherically integrate over the inner cavity surface ('REV uncertainty'). Furthermore, the heat flux plate (HFP) itself adds another resistance to the heat flow from the rock slab interior to the cavity ('resistance error', Hukseflux HFP manual (2016)) which is within $20\%$ and sufficient in our context, given the field conditions and the REV uncertainty. The heat flux plates are cemented onto the block surface to avoid air gaps and to minimise the contact resistance. We assess the resistance error by conceptualizing the heat flow through the block and the HFP into the open cavity as resistances in series. The total resistance $R_{tot}$ is the sum of the conductive resistance in the rock slab within a 'zone of influence' of the HFP given by the sensor diameter $d_{\mathrm{HFP}}$, $R_r = d_{\mathrm{HFP}}/k_r \approx 80 \times 10^{-3}$ m$/2.5$ W m$^{-1}$ K$^{-1}$, the HFP resistance $R_{\mathrm{HFP}} = 71 \times 10^{-4}$ K m$^2$ W$^{-1}$, and the interfacial radiative-convective resistance $R_s$ between the rock surface and the cavity, i.e. $R_{tot} = R_r + R_{\mathrm{HFP}} + R_s$. An upper bound for $R_s$ is the stagnant (no convection), radiation-only inverse heat transfer coefficient $1/h_{rad} = (4\varepsilon\sigma\bar{T}^3)^{-1} \approx 1/5$ K m$^2$ W$^{-1}$ ($\varepsilon \approx 1$). Hence, the HFP is the least resistive link of the heat transfer chain.

## 4 Parameterisations and heat flux modelling

### 4.1 Heat transfer and air circulation

Heat is transferred by three basic modes — *convection/advection* by moving fluid parcels (air/water), the emission/absorption of electromagnetic (thermal infrared) *radiation*, and thermal *conduction* by molecular interaction — whose dominance in unfrozen (excluding latent heat effects) porous materials such as soils under field conditions is controlled by debris texture (characteristic particle size) and water content (degree of saturation) (Johansen, 1975). These heat transfer modes combined result in the ground heat flux $Q_G$ (Eq. 4). In coarse debris far from water saturation, the most important heat transfer modes are *air convection*, heat carried by air circulation, and *thermal radiation* ($Q_r$), radiative heat transfer between blocks of different temperatures by electromagnetic waves that travel across the pore space (Fillion et al., 2011). Heat advection by intercepted rainfall that percolates to the ground-ice table results in a small rain heat flux $Q_{Pr}$ (part of $Q_G$ in Eq. 4). *Heat conduction* alone compared to radiation is considered negligible in the coarse blocky AL because the contact areas between the blocks are too small in the clast-supported debris (cf. Esence et al., 2017), but transfers the heat within the blocks and also in the permafrost body beneath the AL, $Q_{\mathrm{PF}}$ (Scherler et al., 2014; Schneider, 2014). Hence, we conceptualise conductive heat transfer within the blocks and radiative heat transfer between the blocks as a heat transfer chain in series, and denote it as conductive–radiative. We outline a simple heat transfer model in terms of a thermal resistance circuit in Appendix Sect. A.

Furthermore, it is useful to differentiate two types/modes of air convection according to the driving force, (i) buoyancy-driven and (ii) forced convection (Nield and Bejan, 2017). *Buoyancy-driven convection* refers to air set in motion by air density instabilities within the coarse blocky AL, i.e. when denser ($\sim$colder, drier) air is on top of lighter ($\sim$warmer, more moist) air,

and is driven by gravity. The entire unstable air column is set in motion. It exerts a cooling effect: Comparatively colder air sinks into the coarse blocky AL, displaces the warmer air, and subsequently impedes the penetration of warmer air. Warmer air is evacuated rapidly (*Balch effect* or *thermal semiconductor effect*). The vigour of buoyancy-driven convection in a porous medium is a function of the Rayleigh–Darcy number Ra defined as (Nield and Bejan, 2017; Johansen, 1975; Kane et al., 2001; Herz, 2006; Côté et al., 2011; Wicky and Hauck, 2020)

$$\text{Ra} := \frac{\rho_a C_p}{k_{\text{eff}}^0} \frac{g\beta_a K h_{al} \Delta T_a}{(\mu_a/\rho_a)}, \tag{3}$$

where $\rho_a$ is the air density [kg m$^{-3}$], $C_p$ the isobaric specific heat capacity [J kg$^{-1}$ K$^{-1}$], $k_{\text{eff}}^0$ the stagnant (absence of convection) bulk thermal conductivity of the AL [W m$^{-1}$ K$^{-1}$], $\beta_a \approx 1/T_0$ the thermal expansion coefficient [(273 K)$^{-1}$], $K$ the intrinsic AL permeability estimated with the Kozeny–Carman equation [$2 \times 10^{-5}$ m$^2$, Eq. D1] (Herz, 2006; Côté et al., 2011; Wicky and Hauck, 2020), $h_{al}$ the AL layer thickness [m], $\Delta T$ the temperature difference across the AL [K or °C], $\mu_a$ the air dynamic viscosity [Pa s], and $g$ the gravitational acceleration [9.81 m s$^{-2}$]. Buoyancy-driven convection can be expected when Ra exceeds a threshold value (critical Rayleigh number Ra$_c$), commonly given as 27 in open voids or 40 beneath a snow cover. *Wind-forced convection* or *continuous air exchange with the atmosphere* (Humlum, 1997; Harris and Pedersen, 1998; Kane et al., 2001; Juliussen and Humlum, 2008) refers to air set in motion by external wind, i.e. at the ground surface. Wind gusts propagating into the permeable AL (shear flow, momentum diffusion by rough surface) lead to forced mechanical mixing. It can rapidly cool or warm the ground, depending on the air temperature relative to the ground temperature. The mixing is most pronounced near the surface and decays the stronger with depth, the more stable the air density stratification is (Evatt et al., 2015). A labile, (near-)isothermal air column is most easily mixed.

## 4.2 Energy budget of the Murtèl coarse blocky active layer (AL)

The ground heat flux $Q_G$ and the rain heat flux $Q_{Pr}$ [W m$^{-2}$] from the ground surface downwards into the coarse blocky AL are spent on warming the debris $\Delta H_{al}^{\theta}$ (sensible heat storage changes), melting ground ice in the AL $Q_m$ (latent heat storage changes), and conducted into the permafrost body beneath $Q_{\text{PF}}$ ('permafrost heat flux') (cf. Hayashi et al., 2007; Woo and Xia, 1996),

$$(Q_G + Q_{Pr}) - (\Delta H_{al}^{\theta} + Q_m + Q_{\text{PF}}) = 0 \ [\text{W m}^{-2}]. \tag{4}$$

The fluxes are counted as positive if they provide energy to the reference volume, the AL. We use different approaches to independently estimate each term in Eq. 4 (Sect. 4.3) and compare with direct measurements (Sects. 3.2.1, 3.2.3).

## 4.3 Heat flux estimations

### 4.3.1 Ground surface heat flux $Q_G$

The ground heat flux at the surface, $Q_G$, is estimated using the measured thermal net radiation $Q_{\text{CGR3}}^{rad}$ extrapolated to the surface by the transient heat storage in the layer between surface and pyrgeometer depth. Details are in Amschwand et al. (2024a).

### 4.3.2 Rain heat flux $Q_{Pr}$

The flux of infiltrating rainwater $r$ [m$^3$ m$^{-2}$ s$^{-1}$] is the intercepted rainwater (from the rain gauge data) that rapidly percolates to the ground ice table and is cooled from the initial precipitation temperature $T_{Pr}$ to 0°C (at the most) (Sakai et al., 2004; Hayashi et al., 2007). It releases a sensible heat flux

$$Q_{Pr} = C_w r \left(T_{Pr} - 0°\text{C}\right), \quad \text{if } h_S = 0, \, T_{wb} \geq 2°\text{C}, \tag{5}$$

with the volumetric heat capacity $C_w = \rho_w c_w$ [10$^3$ kg m$^{-3}$ × 4.2 kJ kg$^{-1}$ K$^{-1}$], and snow height $h_S$ [m]. As a conservative estimate, the rainwater is assumed to be cooled from surface temperature $T_{Pr} := T_s$ to the freezing point. Precipitation data is taken from the on-site rain gauge, assuming that precipitation is liquid based on a threshold air temperature of $T_{wb} = 2°$C (Amschwand et al., 2024a). Water contributions from upslope flowing onto the rock glacier and liquid precipitation falling into the snowpack is not accounted for (no reliable precipitation measurements from snow-covered rain gauge).

### 4.3.3 Sensible heat storage changes $\Delta H_{al}^\theta$

The sensible heat $\Delta H_{al}^\theta$ stored/released by temperature changes of the blocks are estimated using the calorimetric method based on AL temperatures via (Amschwand et al., 2024a)

$$\Delta H_{al}^\theta = \int_{-h_{al}}^{0} \frac{\mathrm{d}}{\mathrm{d}t}\{(1-\phi_{al})\rho_r c_r T_r(z)\}\,\mathrm{d}z \approx (1-\phi_{al})\frac{\langle \rho_r c_r\rangle}{\Delta t_r}\sum_i\{\langle\bar{T}_a(z_i, t+\Delta t_r)\rangle - \langle\bar{T}_a(z_i,t)\rangle\}\Delta z_i \tag{6}$$

with $h_{al} = 3$ m the AL thickness, $\rho_r$ the rock density [2690 kg m$^{-3}$] (Corvatsch granodiorite, Schneider (2014)), $c_r$ the specific heat capacity [790 J kg$^{-1}$ K$^{-1}$], AL porosity $\phi_{al} = 0.4$ (Scherler et al., 2014), and $T_r(z)$ and $T_a(z)$ the vertical rock and in-cavity air temperature profile [°C], respectively. In practice, we use the AL temperature $T_{al}$ [°C] as measured by the thermistor string TK1 hanging in air (Tables 1, 2) and integrating over timescales where local thermal equilibrium (LTE) holds (discussed in Appendix Sect. A). In the discretised formulation, the temperatures $\langle\bar{T}(z_i)\rangle$ are layer-wise averages in the $i$-th layer with thickness $\Delta z_i$ (denoted by $\langle\cdot\rangle$), derived from the thermistor string TK1/1 and the radiometric surface temperature $T_s$. Since AL water contents are always low enough not to significantly influence the heat capacity, $C_v = (1-\phi_{al})\rho_r c_r$ is a time-invariant, fixed AL thermal property (Scherler et al., 2014).

### 4.3.4 Ground-ice melt heat flux $\text{dev}_{al}$, $Q_m$

The latent heat consumed by melting ground ice is on the one hand estimated from the deviation of the AL budget (Eq. 4), i.e., the residual $\text{dev}_{al} := Q_G + Q_{Pr} - |Q_{PF}|$, and on the other hand using the stake measurements (denoted as $Q_m$) via

$$Q_m = f_i L_m \rho_i \frac{\mathrm{d}\zeta}{\mathrm{d}t}, \tag{7}$$

where $\zeta$ is the observed depth of the ground-ice table [m]. $f_i$ [−], $L_m$ [3.34 × 10$^5$ J kg$^{-1}$], and $\rho_i$ [kg m$^{-3}$] are the volumetric ice content, latent heat of melting, and ice density, respectively.

#### 4.3.5 AL base flux through permafrost body $Q_{\mathrm{PF}}$

The heat flux across the permafrost table $Q_{\mathrm{PF}}$ is estimated with the gradient method from PERMOS borehole temperature data via Fourier's heat conduction equation

$$Q_{\mathrm{PF}} = -k_{\mathrm{PF}}\frac{\mathrm{d}T}{\mathrm{d}z} \approx -k_{\mathrm{PF}}\frac{\Delta T_{\mathrm{PF}}}{\Delta z}, \tag{8}$$

where the borehole temperatures are measured at $4$ and $5\,\mathrm{m}$ depth in the permafrost body beneath the AL. We take a constant thermal conductivity $k_{\mathrm{PF}}$ value of $2.5\,\mathrm{W\,m^{-1}\,K^{-1}}$ (Vonder Mühll and Haeberli, 1990; Scherler et al., 2014).

### 4.4 Stefan parameterisation of ground-ice melt

If the ground heat flux is mostly spent on melting ground ice (discussed in Sect. 6.1), the rate of lowering the thaw front/ground-ice table $\mathrm{d}\zeta/\mathrm{d}t\,[\mathrm{m\,s^{-1}}]$ can be approximated with a linearised heat conduction equation (Hayashi et al., 2007),

$$\rho_i f_i L_m \frac{\mathrm{d}\zeta}{\mathrm{d}t} = k_{\mathrm{eff}}\frac{T_s - 0°\mathrm{C}}{\zeta}, \tag{9}$$

whose solution is the Stefan equation of the form (Hayashi et al., 2007)

$$\zeta(t) = \sqrt{\frac{2k_{\mathrm{eff}}I(t)}{\rho_i f_i L_m}}, \tag{10}$$

where $k_{\mathrm{eff}}$ is the effective thermal conductivity of the (unfrozen) AL above the thaw front/ground-ice table $[\mathrm{W\,m^{-1}\,K^{-1}}]$ located at depth $\zeta\,[\mathrm{m}]$ beneath ground surface (Fig. 4), and $I(t)$ the surface thaw index $[°\mathrm{C}\times\mathrm{d}]$ (defined below). Two modifications are necessary to account for the AL stratigraphy on Murtèl (Fig. 4). First, the seasonal lowering of the ground ice table (assumed to coincide with the thaw front) is modelled with a modified Stefan equation for a two-layered AL (Nixon and McRoberts, 1973; Kurylyk, 2015). In Eq. 10, the frozen ground is initially (at the onset of thaw season) uniform. However, on Murtèl, the ice does not fill the AL pore space up to the ground surface. Rather, a layer (with thickness $h_1$) on top of the ice-saturated AL remains nearly ice free year-round. This ice-poor overburden dampens ground-ice melt rates/thaw rates from the onset of the thaw season. Apart from the different ground-ice content, the two layers share the same properties (porosity). The second modification is a correction factor $\lambda_5$ for sensible heat storage changes in the thawed AL. $\lambda_5$ is derived from the Stefan number Ste, $\lambda_5 = 1 - 0.16\,\mathrm{Ste} + 0.038\,\mathrm{Ste}^2$ (Kurylyk and Hayashi, 2016). The depth-averaged dimensionless Stefan number Ste is proportional to the ratio of sensible heat to latent heat absorbed during thawing (Kurylyk and Hayashi, 2016),

$$\mathrm{Ste} := \frac{C_v \bar{T}_s}{L_m \langle f \rangle \rho_i}, \tag{11}$$

with the bulk volumetric heat capacity $C_v = (1 - \phi_{al})\rho_r c_r\,[\mathrm{J\,m^{-3}\,K^{-1}}]$ of the (unfrozen, ice-free) AL (identical for both layers), the average surface temperature $\bar{T}_s$ for the time $t$ elapsed since onset of the thaw season, and the latent heat consumed by the melting of the ground ice $L_m \langle f \rangle \rho_i$ (different in each layer and depth-averaged denoted by $\langle \cdot \rangle$; details in Kurylyk and Hayashi (2016)). In other words, the Murtèl AL stratigraphy calls for a multi-layer Stefan equation accounting for sensible

heat storage. We use the extension of Eq. 10 developed by Kurylyk (2015) and Aldrich and Paynter (1953) to predict the thaw depth $\zeta$, which for our purpose reduces to (because of $k_1 = k_2 := k_{\text{eff}}$ in Eq. 23 in Kurylyk, 2015)

$$\zeta(t) = \begin{cases} \sqrt{\frac{2k_{\text{eff}}I_t(t)}{L_m f_1 \rho_i}}, & \text{if } I_t(t) \leq I_1 \text{ (ice-poor overburden)} \\ \sqrt{h_1^2 \left(1 - \frac{f_1}{f_2}\right) + \frac{2k_{\text{eff}}I_t(t)}{L_m f_2 \rho_i}}, & \text{if } I_t(t) > I_1 \text{ (ice-rich layer)}, \end{cases} \tag{12}$$

with the effective thermal conductivity $k_{\text{eff}}$ [W m$^{-1}$ K$^{-1}$] of the (thawed/unfrozen) AL above the thaw front/ground-ice table, the volumetric pore ice content $f$ of each layer (maximum at saturation, $f \leq \phi_{al}$), and the pore ice density $\rho_i$ [900 kg m$^{-3}$]. The total surface thaw index $I_t(t)$ [°C $\times$ d] is defined as

$$I_t(t) := \int_0^t \lambda_5^2 T_s(t') \, dt'. \tag{13}$$

and the thaw index of the ice-poor AL overburden $I_1$ as

$$I_1 := \frac{h_1^2 L_m f_1 \rho_i}{2k_{\text{eff}}}. \tag{14}$$

Eq. 12 can be reduced if we assume that the ice-poor overburden has a negligible ice content, $f_1 \ll f_2$, and approximates the thaw depth late in the thaw season when $I_t(t) \gg I_1$. Then, Eq. 12 reduces to

$$\zeta(t) = \sqrt{h_1^2 + \frac{2k_{\text{eff}}I_t(t)}{L_m \rho_i (f_2 - f_1)}} \overset{f_2 \gg f_1}{\approx} \sqrt{h_1^2 + \frac{2k_{\text{eff}}I_t(t)}{L_m \rho_i f_2}} \quad \text{for } I_t(t) > I_1. \tag{15}$$

Eq. 15 is an useful approximation when the overburden is thick and ice poor, i.e. the effect of $f_1$ is small compared to the effect of $h_1$. Without the damping effect of the overburden ($h_1 \rightarrow 0$), Eq. 15 reduces further to the "classic" Stefan solution (Eq. 10).

In discretised form with daily average surface temperatures, $I_t(t_i) = 86400 \sum_i (\bar{\lambda}_5^2 \bar{T}_s)[t_i]$, summed over the $i$-th day since onset of the thaw season (Hayashi et al., 2007). Eq. 12 is premised on the assumption of (i) initial uniform temperature at the freezing point throughout the AL and uniform temperature beneath the thaw front (zero heat flux from the permafrost body beneath; Sect. 6.1), (ii) layer-wise homogeneous and time-invariant thermal properties and ground ice content, and (iii) quasi-steady state conditions (Kurylyk and Hayashi, 2016).

### 4.5 Estimation of AL heat transfer parameters ($k_{\text{eff}}$, $\kappa_a$)

We derive the heat transfer parameters of the AL, the effective thermal conductivity $k_{\text{eff}}$ and the apparent thermal diffusivity $\kappa_a$, from in-situ measurements using two different approaches based on different measurements. $k_{\text{eff}}$ and $\kappa_a$ are related via the volumetric heat capacity $C_v$ (Vonder Mühll and Haeberli, 1990),

$$\kappa_a = \frac{k_{\text{eff}}}{C_v} = \frac{k_{\text{eff}}}{(1 - \phi_{al})\rho_r c_r}. \tag{16}$$

Values for the porosity $\phi_{al}$, rock density $\rho_r$, and the heat capacity $c_r$ given in Sect. 4.3.3 yield $C_v = 1.275$ MJ m$^{-3}$ K$^{-1}$.

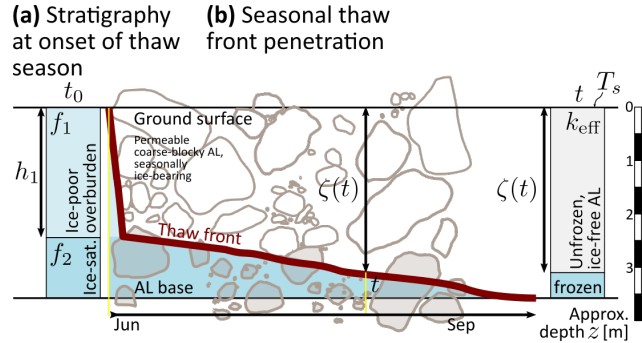

**Figure 4.** Ground-ice thaw and the Stefan equation. **(a)** Initial stratigraphy at the onset of the thaw season with ice-poor overburden and ice-saturated layer. **(b)** Seasonal thaw front penetration.

### 4.5.1 Effective thermal conductivity $k_{\text{eff}}$ estimation

The effective thermal conductivity $k_{\text{eff}}$ [W m$^{-1}$ K$^{-1}$] is derived from the measured AL thermal radiation $Q_{\text{CGR3}}^{rad}$ and vertical AL temperature gradient $\frac{\mathrm{d}T_{al}}{\mathrm{d}z} := \nabla_z T_{al}$ using a diffusive flux-gradient relation of the form

$$Q_{\text{CGR3}}^{rad} = -k_{\text{eff}} \frac{\mathrm{d}T_{al}}{\mathrm{d}z}. \tag{17}$$

This approach yields a thaw-season averaged effective thermal conductivity $k_{\text{eff}}$ [W m$^{-1}$ K$^{-1}$], that we denote by $\bar{k}_{\text{eff}}^{rad}$ for precision. It might seem odd to use Eq. 17, a diffusion equation formally identical to Fourier's heat conduction equation (cf. Eq. 8), since heat conduction is considered insignificant in the coarse blocky AL (Sect. 4.1; discussed in Sect. 6.3.2). However, radiative heat transfer in a porous medium with opaque particles (rock) and transparent fluid (air) can be expressed as diffusive (Fillion et al., 2011; Lebeau and Konrad, 2016). $\bar{k}_{\text{eff}}^{rad}$ is an effective thermal conductivity that lumps conductive/radiative and non-conductive (convective) heat transfer in the highly permeable blocky material (Herz, 2006).

### 4.5.2 Apparent thermal diffusivity $\kappa_a$ estimation

The apparent thermal diffusivity $\kappa_a$ [m$^2$ s$^{-1}$] is derived from measured AL temperatures $T_{al}$ using the derivative method based on the one-dimensional transient thermal diffusion equation (Biot–Fourier equation) (Hinkel et al., 1990; Conway and Rasmussen, 2000),

$$\frac{\mathrm{d}T_{al}}{\mathrm{d}t} = \kappa_a \frac{\mathrm{d}^2 T_{al}}{\mathrm{d}z^2}, \tag{18}$$

where $t$ [s] is the time, and $z$ is the AL depth [m]. Analogous to $\bar{k}_{\text{eff}}^{rad}$, $\kappa_a$ is an apparent thermal diffusivity that lumps conductive/radiative and non-conductive (convective) heat transfer. Only the latent effects of freezing/thawing is minimized by using values from the unfrozen AL ($T_{al} > 0.5°$C). Day-to-day temperature change $\mathrm{d}T_{al}/\mathrm{d}t$ and the second derivative $\mathrm{d}^2 T_{al}/\mathrm{d}z^2$ are calculated using the Petersen et al. (2022) algorithm from daily average AL temperature data (TK1/2–4). To avoid spurious $\kappa_a$ values, no $\kappa_a$ is calculated for near-isothermal conditions (unstable numerics; Hinkel et al., 1990).

 **5   Measurement results**

## 5.1   Meteorological conditions

The weather in each season differed markedly in the two years 2020–2022, which is reflected by air temperature and snow at
the surface (Fig. 5) and the AL temperatures at depth (Fig. 6a). Winter 2020–2021 was colder ($-6.2^\circ$C Nov–April average),
more snow-rich (120 cm Feb peak measured on a windswept ridge) and lasted longer than the unusually snow-poor (two 80 cm
peaks in Dec and Apr), short winter 2021–2022 with snow disappearance in May–June, one month earlier than the usual melt-
out in July. AL temperatures fluctuated more and attained lower values ($T_{al} < -8^\circ$C) in less cold ($-5.3^\circ$C) winter 2021–2022.
Summer 2021 was cool-wet compared to the hot–dry summer 2022; temperatures were lower (July–August: average: $6.9^\circ$C vs.
$9.3^\circ$C) with frequent passage of weather fronts, often bringing cold air ($\leq 3^\circ$C; minimum daily average temperature: $0.7^\circ$C vs.
$5.6^\circ$C) and mixed precipitation (sleet). A few snow patches survived in the Murtèl catchment, which has rarely been occurring
in the last $\sim 15$ years (M. Hoelzle and C. Hauck, pers. comm.). In contrast, the summer 2022 was marked by heat waves
($T_{al} > 10^\circ$C in June, July) co-occurring with dry spells. The surface meteorological conditions are described in more detail in
Amschwand et al. (2024a).

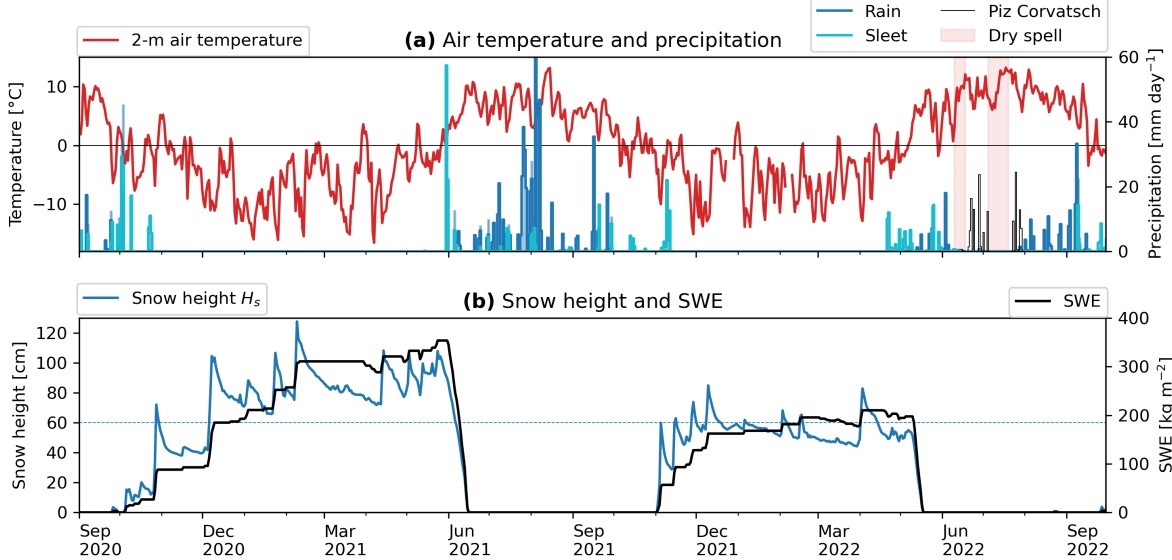

**Figure 5.** Meteorological conditions. **(a)** Air temperature (daily mean) and precipitation (daily sum). **(b)** Snow height and SWE. Rain and
sleet (mixed precipitation) separated based on a wet-bulb temperature threshold of $2^\circ$C (Amschwand et al., 2024a). A snow height of $h_S =$
60 cm (measured on a windswept ridge) discriminates between a semi-closed and a closed/insulating snowpack (cf. Fig. 8). Precipitation
data at station *Piz Corvatsch* from MeteoSuisse.

## 5.2 Ground thermal and moisture regime

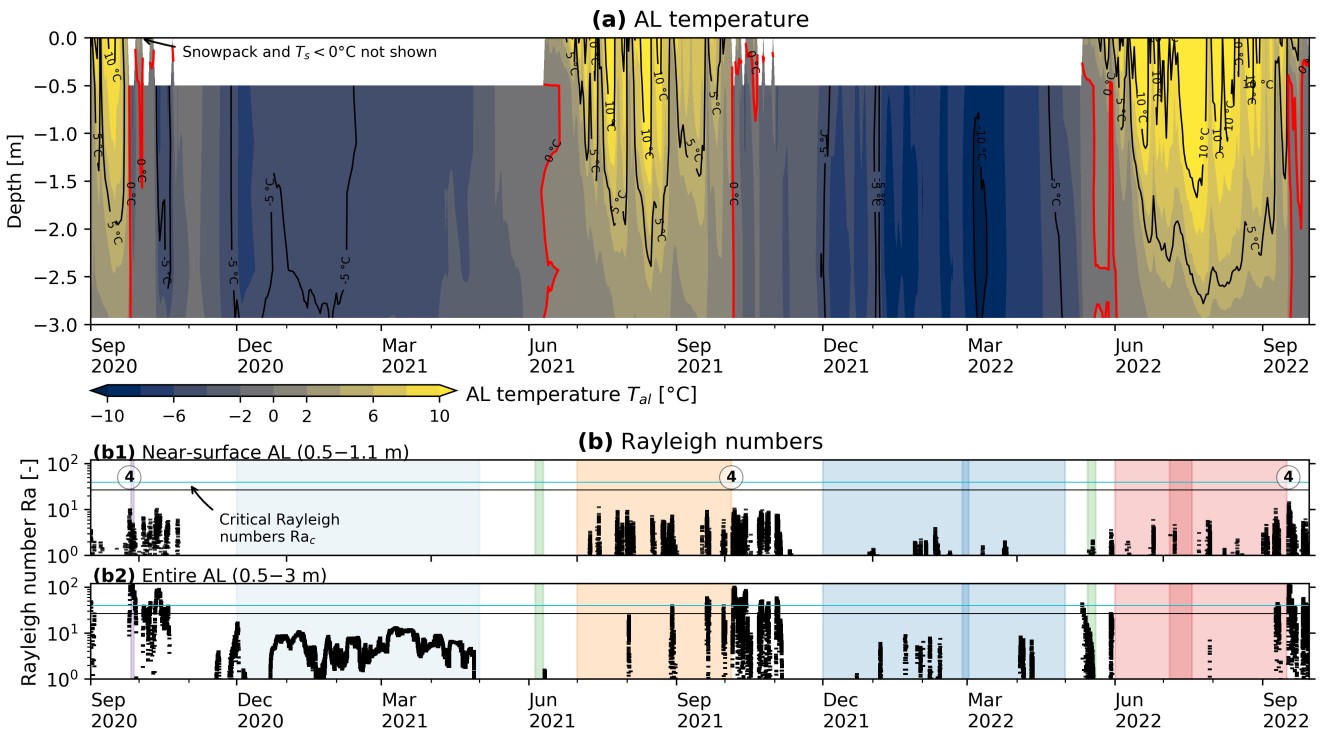

**Figure 6. (a)** AL temperatures $T_{al}$ in the instrumented main cavity (contour plot from TK1 data). Selected isotherms shown as contour lines. **(b)** Rayleigh numbers Ra indicate the air column stability in the near-surface AL **(b1)** and entire AL **(b2)** (Eq. 3, 10 minute resolution). Coloured periods are those shown in Fig. 7. Circled number ④ refers to Table 3.

We characterize the ground thermal regime in terms of the Rayleigh–Darcy number Ra (Eq. 3). Ra numbers indicate the
365 stability of the AL air column in a shallow sub-layer ($0.5−1.1$ m, Fig. 6b1) and the entire AL ($0.5−3$ m, Fig. 6b2). Onset of buoyancy-driven convection is potentially at Ra $\geq$ Ra$_c$ of 27 in snow-free and 40 in snow-covered conditions. Autumn 2020 starts at the end of Sep with rapid cooling at high, supercritical Rayleigh numbers. Cooling continues more slowly throughout November, until a thick, 'closed' snow cover stalled winter cooling in Dec 2020, when the snow height exceeded a threshold ($60$ cm on a rock glacier ridge; Fig. 5b). The AL remained near-isothermal and near-isohume at sub-critical Ra numbers until
May 2022. Summer 2022 was characterized by frequent shallow instabilities, i.e. super-critical Ra numbers in the uppermost $1$ m of the AL. The deep AL remained stably stratified. The instabilities became more frequent and encompassed the entire AL in Oct 2021. In the snow poor winter 2021–2022, the snow cover remained 'open' as the snow height only rarely exceeded the threshold. AL temperature kept fluctuating at occasionally sub-critical Ra numbers. In summer 2022, shallow instabilities occurred less frequently than in summer 2021 (Fig. 6b1).

**Table 3.** Typical temperature profiles, heat fluxes and airflow patterns during different seasons, characteristic weather patterns, and snowpack conditions on Murtèl rock glacier (ciphers ①–⑥ are referred to in the text and figures).

| # | Condition | Temperature profile and Rayleigh number | Heat flux (daily average) | Air circulation mode |
|---|---|---|---|---|
| *Processes on seasonal timescale* (over weeks–months) | | | | |
| ① | Thaw season | Mostly[a] near-linear profile (daily timescale), positive gradient (stable) | $Q_{\mathrm{HFP}} \propto Q_{\mathrm{CGR3}}^{rad} \propto \nabla_z T_{al}$, $5-15$ W m$^{-2}$ downwards, $Q_{\mathrm{CGR3}}^{rad} \propto T_s$ | *Wind-forced convection* enhances radiative–conductive heat transfer |
| ② | Winter stagnant/ closed snow cover | Near-linear profile, isothermal or weakly unstable gradient, slowly evolving | $Q_{\mathrm{HFP}} \approx Q_{\mathrm{CGR3}}^{rad}$, $< 2$ W m$^{-2}$ upwards | Calm/stagnant, no convective AL–atmosphere coupling |
| ③ | Winter semi-closed snow cover | 'Bulged' and fluctuating profile | $Q_{\mathrm{HFP}} > Q_{\mathrm{CGR3}}^{rad}$, often anti-correlated, $2-10$ W m$^{-2}$ upwards | *Cold-air infiltration* through semi-closed snow cover (snow funnels) |
| *Short-lived events* (hours–days) | | | | |
| ④ | Convective overturning | Unstable ($\mathrm{Ra} > \mathrm{Ra}_c$), rapidly changing (transient) | $Q_{\mathrm{HFP}} \gg Q_{\mathrm{CGR3}}^{rad}$, large: $20-30$ W m$^{-2}$ upwards | *Rayleigh ventilation* (dominant heat transfer mode) |
| ⑤ | Storm-wind mixing | Strongest if (near-)isothermal | Small (minor impact[b]) | Wind-forced convection |
| ⑥ | Water refreezing | Rapid temperature rise towards $0°$C at all AL depths beneath ripe ("warm") snowpack | $Q_{\mathrm{HFP}} > Q_{\mathrm{CGR3}}^{rad}$, $4-8$ W m$^{-2}$ downwards | Typically calm (minor impact) |
| | | Figs. 6b, 7 | Figs. 8, 9, A1 | Figs. 11, 15, B1, E2 |

[a] Exceptions are dry-hot weather spells (Fig. 7a) or during the passage of weather fronts (Rayleigh ventilation). [b] In wind-sheltered Murtèl cirque.

The temperature profiles are specific to certain conditions that we mark with circled numbers ①–④ and use throughout this work. An overview is given in Table 3 which shows how the heat transfer processes are reflected by the below-ground temperature and airflow measurements. The thaw season temperature profiles (Fig. 7a, ①) are near-linear down to daily timescale, but not on sub-diurnal timescales or on days with rapid cooling (Rayleigh ventilation ④, e.g., during the passage of cold fronts). Thaw-season average temperature gradients are 2.0 K m$^{-1}$ for 2021 ($R^2 = 0.995$) and 2.8 K m$^{-1}$ for 2022 ($R^2 = 0.998$). The near-surface AL is often warmer and therefore unstable with respect to the atmosphere despite stable air stratification inside the AL (nonlocal static stability sensu Stull (1991)). The winter averages (Dec–Apr) are near-isothermal in winter 2020–2021 (②). In winter 2021–2022, occasional temperature minima at roughly 2 m depth ('bulges') hint at lateral cold air flow (Sect. 6.4, ③). The locally (near-)stable AL air is nonlocally unstable compared to the radiatively cooled snow surface with average surface

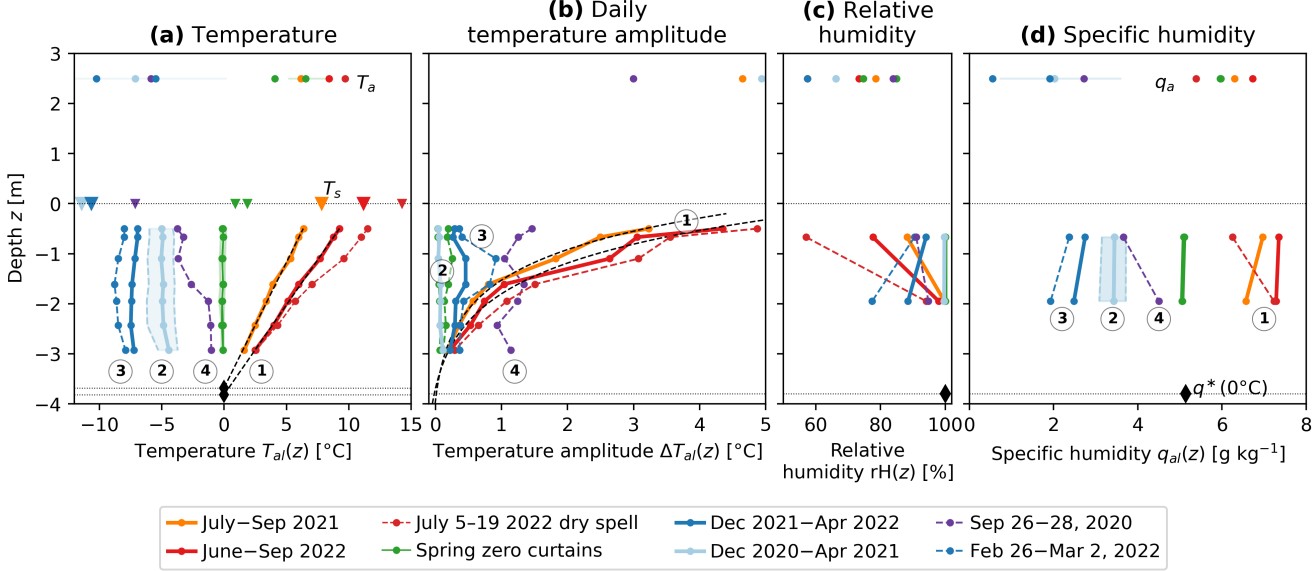

**Figure 7.** Characteristic temperature and humidity profiles. **(a)** Vertical temperature profiles ('trumpet curves') during selected periods (the circled numbers ①–④ refer to Table 3): ① summer/thaw season; ② stagnant winter conditions (Dec 2020–Apr 2021); ③ winter-time cold-air infiltration (winter average Dec 2021–Apr 2022, infiltration period Feb 26–March 2); ④ convective overturning by Rayleigh ventilation (event of Sep 26–28, 2020). **(b)** Amplitude of daily temperature variation (max–min). On a timescale of one day or longer, the thaw-season temperature profiles are near-linear and daily amplitude decays exponentially with depth, even in the comparatively large instrumented cavity. **(c, d)** Vertical relative rH and specific $q_{al}$ humidity profiles.

temperature $T_s$ of $-10°$C (but can go as low as $-30°$C). Note the striking asymmetry between near-isothermal winter and steep ($2.0-2.8$ K m$^{-1}$) thaw-season temperature profiles (asymmetric envelopes).

The temperature amplitude $\Delta T$ is attenuated exponentially with depth $z$ (Fig. 7b) proportional to $\exp\{-1.083z\}$. Looking at sub-daily resolution, the AL temperature showed a daily course without time lag down to $-2.9$ m, only with attenuated amplitudes (Appendix Fig B1a–b). Specific humidity gradients (Fig. 7d) averaged over days–weeks were parallel to the temperature gradients because the AL is most often close to saturation (Fig. 7c). Exceptions were the summer 2022 dry spells that dried out the AL.

### 5.3 Heat flux plate and pyrgeometer measurements

The direct heat flux plate $Q_{\text{HFP}}$ and pyrgeometer $Q_{\text{CGR3}}^{rad}$ (thermal infrared radiation) measurements give an overview of flux magnitudes and seasonality (Fig. 8). For clarity, we describe the heat fluxes at seasonal down to daily resolution. Sub-daily (hourly) resolution data provide additional insights discussed in Appendix Sect. B. The measured heat fluxes vary seasonally primarily according to the snow conditions and the AL temperature gradient $\nabla_z T_{al}$ (characteristic temperature profiles are

shown in Fig. 7). The circled numbers ①–⑥ refer to the characteristic weather conditions introduced in Fig. 7 and are marked in Figs. 6b, 8, and 11 (Table 3).

During the thaw season ($T_{al} > 0°C$), when temperature gradients are most often stable ($\nabla_z T_{al} > 0$ K m$^{-1}$), measured daily average heat fluxes $Q_{\mathrm{HFP}}$ and $Q_{\mathrm{CGR3}}^{rad}$ are $5-20$ W m$^{-2}$ downwards (①). The downward ($Q_{\mathrm{HFP}} > 0$) flux into the block where the HFP/1 is placed on (warming, positive sign) is strongly correlated ($R^2 = 0.9$) with the net downward radiation in the instrumented cavity ($Q_{\mathrm{CGR3}}^{rad} > 0$) (Fig. 8, ①), with $Q_{\mathrm{HFP}} = 0.7 Q_{\mathrm{CGR3}}^{rad} + 0.7$ W m$^{-2}$ (Appendix Fig. A1). In other words, $Q_{\mathrm{HFP}} > 0$ is congruent with $0.7 Q_{\mathrm{CGR3}}^{rad}$ (histogram in Fig. 8b). The remaining deviation defined as $Q_{\mathrm{HFP}} - 0.7 Q_{\mathrm{CGR3}}^{rad}$ is generally insignificant (within $\pm 2$ W m$^{-2}$) during the thaw season, suggesting that radiative–conductive heat transfer dominates: $Q_{\mathrm{CGR3}}^{rad}$ explains the total heat transfer $Q_{\mathrm{HFP}}$ measured by the heat flux plate, $Q_{\mathrm{CGR3}}^{rad} \sim Q_{\mathrm{HFP}}$. The different scaling of the $Q_{\mathrm{HFP}}$ and $Q_{\mathrm{CGR3}}^{rad}$ measurements can be explained by (i) the instrumental uncertainty (notably the $Q_{\mathrm{HFP}}$ resistance error of up to $20\%$), and (ii) the REV uncertainty: The HFP/1 measures the heat flux locally whereas each pyrgeometer integrates hemispherically (with a cosine response) over the cavity surface.

In contrast, non-radiative heat fluxes dominate the upward $Q_{\mathrm{HFP}} < 0$ W m$^{-2}$ fluxes (cooling) in autumn and winter. $Q_{\mathrm{HFP}} < 0$ is congruent with the deviation (histogram in Fig. 8b) but is unrelated to $Q_{\mathrm{CGR3}}^{rad}$, $Q_{\mathrm{CGR3}}^{rad} \not\propto Q_{\mathrm{HFP}}$. Unlike $Q_{\mathrm{CGR3}}^{rad}$, upward $Q_{\mathrm{HFP}}$ fluxes rapidly increase with unstable (negative) AL air temperature gradients (shown by the colors in Fig. A1a), pointing at air convection as the dominant heat transfer process. Additionally, the snowpack modulates the winter-time heat transfer: In winter 2021–2022, heat fluxes were small ($\leq 2$ W m$^{-2}$; ②) beneath a closed snow cover when the snow height $h_S$ exceeds 60 cm (measured on a windswept rock-glacier ridge). In the snow-poor winter 2021–2022, $Q_{\mathrm{HFP}}$ and $Q_{\mathrm{CGR3}}^{rad}$ are episodically anti-correlated, i.e., the downward net radiation increased with rapid cooling (Dec 2021–Feb 2022, ③). Strong cooling occurred in summer and in autumn (Sep–Oct 2020, Oct 2021, Sep 2022; ④) during the passage of cold weather fronts.

In spring (before the onset of the zero curtain) and occasionally also during winter (e.g., Nov 2020), brief events of heat input lead to rapid AL warming towards (but not exceeding) $0°C$. These 'warming spikes' ⑤ often occur during winter/spring 'heat waves' ($T_a > 0°C$), pointing at AL warming by the refreezing of snowmelt via the released latent heat. The attribution is most reliable in case of a closed/insulating snow cover (e.g. in June 2021) when (storm) wind pumping ⑥ can be excluded.

## 5.4 AL heat transfer parameters ($k_{\mathrm{eff}}$, $\kappa_a$)

### 5.4.1 Thaw-season average $k_{\mathrm{eff}}$

During the thaw seasons, daily average thermal net radiation $Q_{\mathrm{CGR3}}^{rad}$ is strongly correlated with the vertical AL air temperature gradient $\nabla_z T_{al}$ in the cavity, although the cavity is small enough that the air in the cavity is transparent to thermal radiation and does not participate in the radiative heat transfer (discussed in Appendix Sect. A). Linear regressions of daily average values of $Q_{\mathrm{CGR3}}^{rad}$ and $\nabla_z T_{al}$ yield $Q_{\mathrm{CGR3}}^{rad} = 3.0 \nabla_z T_{al} + 0.26$ W m$^{-2}$ ($R^2 = 0.957$) for summer 2021, and $Q_{\mathrm{CGR3}}^{rad} = 3.8 \nabla_z T_{al} - 0.45$ W m$^{-2}$ ($R^2 = 0.965$) for summer 2022 (Fig. 9), suggesting $\bar{k}_{\mathrm{eff}}^{rad} = 3.0$ W m$^{-1}$ K$^{-1}$ (for 2021) and 3.8 W m$^{-1}$ K$^{-1}$ (for 2022). These $\bar{k}_{\mathrm{eff}}^{rad}$ values refer to the timescale of an entire thaw season (hence the overbar in $\bar{k}_{\mathrm{eff}}^{rad}$). At that timescale, radiative–conductive heat transfer dominates, but it does not exclude convective heat transfer altogether. In fact, convection

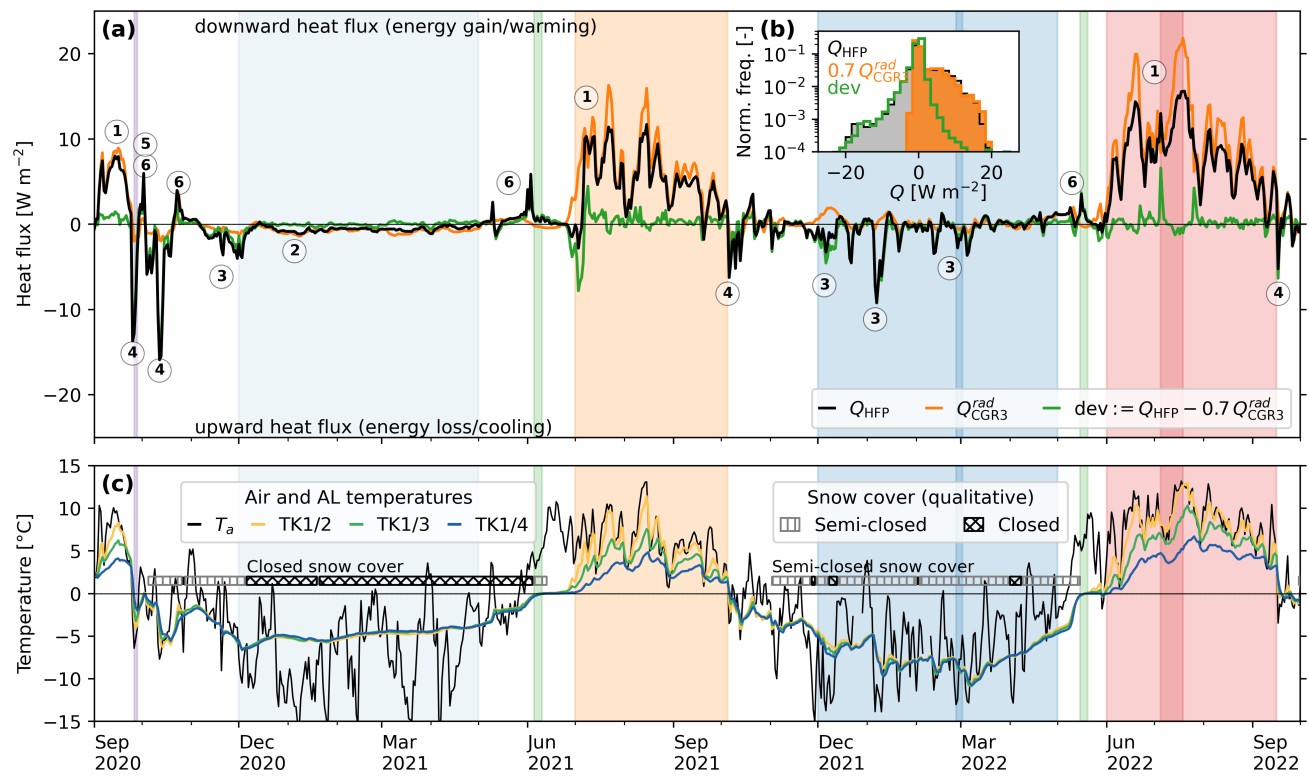

**Figure 8. (a)** Heat flux measured by the heat flux plate HFP/1 at the cavity floor $Q_{\mathrm{HFP}}$, AL thermal net radiation measured by the pyrgeometer pair $Q_{\mathrm{CGR3}}^{rad}$, and the deviation $Q_{\mathrm{HFP}} - 0.7 Q_{\mathrm{CGR3}}^{rad}$ (daily averages). Positive flux is downwards into the rock slab. The circled numbers ①–⑥ refer to Table 3 and are detailed in the text. The snow cover is classified as 'semi-closed' (convective exchange through snow funnels) or 'closed' (no convective AL–atmosphere exchange; Fig. 5b) (Amschwand et al., 2024a, Fig. 4). Inset **(b)** Normalized histogram of the daily average fluxes. Downward fluxes (positive) are mainly conductive/radiative ($Q_{\mathrm{HFP}} > 0$ is congruent with $0.7 Q_{\mathrm{CGR3}}^{rad}$; Fig. A1a), upward fluxes (negative) are mainly non-conductive/convective ($Q_{\mathrm{HFP}} < 0$ is congruent with the deviation). **(c)** Air and AL temperatures and snow cover status (closed/semi-closed). Coloured periods are those shown in Fig. 7.

does occur and appears in the measurements when zooming in to sub-daily resolution. Plotting hourly values of a clear summer day (July 15, 2022, as an example in Fig. 9) reveals clockwise "loops" caused by diurnal cycles of $Q_{\mathrm{CGR3}}^{rad}$ out of phase with $\nabla_z T_{al}$ (black points, midnight value marked by the red cross). AL air temperature leads and thermal net radiation follows (discussed in Appendix Sect. A). Convection that contributes to the total heat transfer to a different extent likely explains the slightly different $Q_{\mathrm{CGR3}}^{rad} - \nabla_z T_{al}$ relation for the two thaw seasons 2021 and 2022. The independently calculated $\kappa_a$ shows the influence of convection on the AL heat transfer parameter more clearly (Sect. 5.4.2).

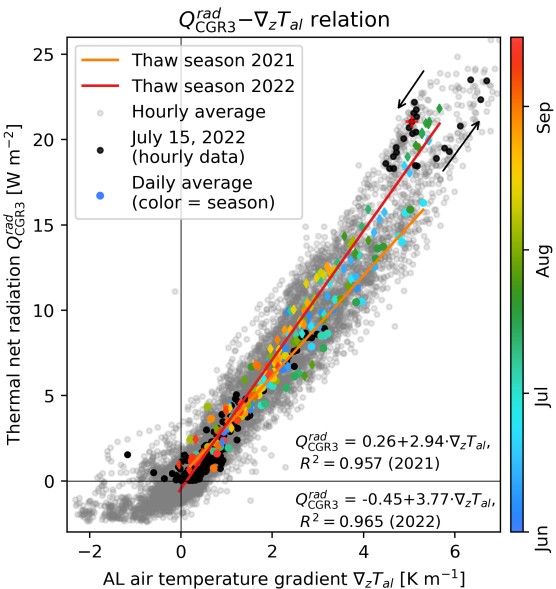

**Figure 9.** Thermal net radiation $Q_{\text{CGR3}}^{rad}$ vs. vertical AL air temperature gradient $\nabla_z T_{al}$. Daily averages are highly correlated (Eq. 17). Hourly values show a diurnal loop.

### 5.4.2  Daily apparent thermal diffusivity $\kappa_a$

The apparent thermal diffusivity $\kappa_a$ (calculated from daily AL temperatures as outlined in Sect. 4.5.2, Eq. 18) during the 2021 and 2022 thaw seasons varies over two orders of magnitude between $2 \times 10^{-5}$ and $2 \times 10^{-7}$ m$^2$ s$^{-1}$ and includes negative values (Fig. 10). $\kappa_a$ systemically varies primarily with the AL temperature gradient $\nabla_z T_{al}$ and secondarily with the atmospheric wind speed $u$. On daily timescale, the impact of convective heat transfer on the heat transfer parameters appears. $\kappa_a$ is largest at unstable or near-isothermal air stratification ($1.9 \times 10^{-5}$ m$^2$ s$^{-1}$ at $\nabla_z T_{al} < 0.5$ K m$^{-1}$), has the largest scatter at weakly–moderately stable conditions ($0.5$ K m$^{-1} < \nabla_z T_{al} < 4$ K m$^{-1}$), and approaches $\kappa_a^0 = 9.6 \times 10^{-7}$ m$^2$ s$^{-1}$ at strongly stable air stratification ($\nabla_z T_{al} > 4$ K m$^{-1}$), where turbulence is suppressed and convective heat transfer is minimal. The thaw-season log-mean $\bar{\kappa}_a$ is $2.3 \times 10^{-6}$ m$^2$ s$^{-1}$, which, converted to $k_{\text{eff}}$ via Eq. 16, yields $2.9$ W m$^{-1}$ K$^{-1}$ that agrees with the independently estimated $\bar{k}_{\text{eff}}^{rad}$. Importantly, the simple explanation of an "insulating" AL in the literal sense of a low thermal conductivity falls short on Murtèl rock glacier: $\bar{k}_{\text{eff}}^{rad}$ is that of the local bedrock (Schneider et al., 2012) or the underlying permafrost body (Weber and Cicoira, 2024), and roughly $10\times$ higher than that of the snowpack (Amschwand et al., 2024a).

### 5.5  Sub-surface airflow

The same three variables that control $\kappa_a$ (Fig. 10), the snow cover, AL temperature gradient $\nabla_z T_{al}$, and atmospheric wind speed $u$, also control the below-ground airflow speeds. Airflow speeds increase with negative $\nabla_z T_{al}$ and increasing $u$ (Fig. E2a) as

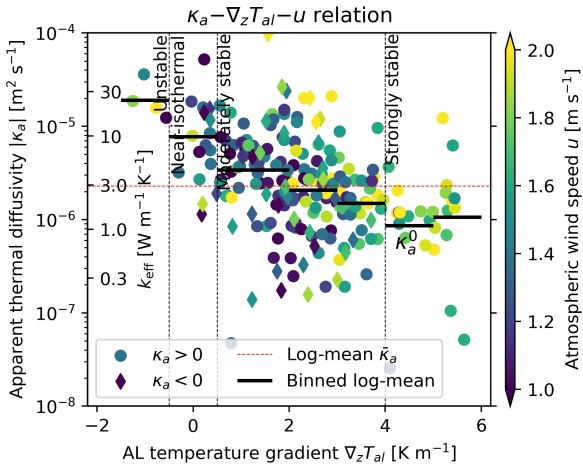

**Figure 10.** Apparent thermal diffusivity $\kappa_a$ during the two thaw seasons 2021 and 2022 ($T_{al} > 1°C$) calculated from daily average AL temperatures in the instrumented cavity (TK1/3 at $1.6$ m depth; Eq. 18).

does $\kappa_a$ (Fig. 10). Hence, the independent airflow speed measurements testify the importance of convection. Airflow speeds (Fig. E1) differ seasonally in terms of (i) vertical airflow speed profile (depth of maximum speed) and (ii) temporal pattern (timing of diurnal oscillations). Fig. 11 shows a data illustration of three characteristic air circulation modes that occurred in the days around the transition from thaw season to autumn 2020, (i) wind-forced shallow ventilation of the stably stratified AL air column, (ii) buoyancy-driven Rayleigh ventilation, and (iii) wind-forced mixing of the isothermal, labilized air column ('storm-wind mixing'). First, during the snow-free thaw season season with unresisted AL–atmosphere connectivity, below-ground airflow follows a strong, regular diurnal cycle with an afternoon speed peak and calm nights (Fig. 11b), in phase with insolation, the surface temperature $T_s$ and the thermally driven (anabatic) local slope winds (Amschwand et al., 2024a) (Fig. B1). This diurnal pattern is shared by all wind speed sensors. Airflow speeds are everywhere highest near the surface (up to $20$ cm s$^{-1}$) and decrease with depth (cf. Evatt et al. (2015)), except in the deeper parts of the instrumented main cavity. There, the lowermost WS/1 mounted in a narrow constriction (Fig. 3, Table 2) showed higher wind speeds and responded more sensitively than the WS/2 in the more spacious mid-cavity. This is however due to the Venturi effect and does not detract from the general observation that wind-forced ventilation under stable temperature gradients decays with depth. At typical depths of the ground-ice table ($3-5$ m, airflow speeds were low (close to the resolution limit of a few cm s$^{-1}$) but tendentially higher under strong atmospheric winds. Hence, the effect of wind-forced convection under stable AL temperature gradients is weak but detectable down to $3$ m depth in the spacious instrumented cavity (Fig. E2a). Second, wind most efficiently mixes an isothermal, labilized AL air column as occurred for example in October 2020 (Fig. 11). This 'storm-wind mixing' had little impact on Murtèl's ground thermal regime because the AL is rarely isothermal under snow-free conditions. Third, Typically in autumn, Rayleigh ventilation under unstable temperature gradients sets the entire air column in motion and leads to rapid cooling of the entire AL within hours–days, for example in late September 2020 (Figs. 6b, 11).

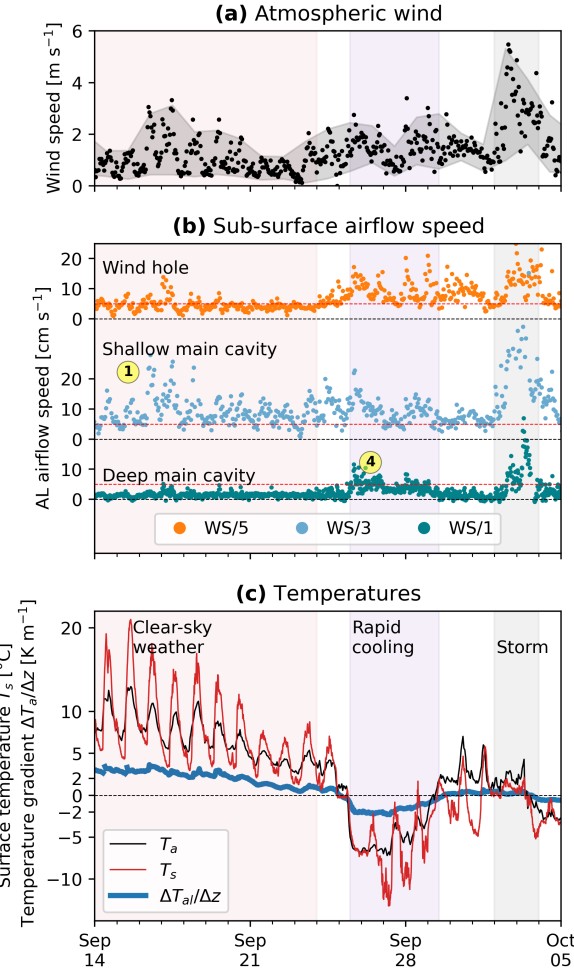

**Figure 11.** Zoom-in to the autumnal cooling in 2020 that illustrates the different air circulation modes in the coarse blocky AL. Above-surface meteorological conditions (insolation and surface temperature, wind speed) and the ground thermal regime (vertical temperature gradient) interact to produce characteristic air circulation modes. **(a)** Atmospheric wind speed. **(b)** Strong diurnal surface heating with shallow *wind-forced ventilation* (WS/3) is characteristic for clear summer days (①). Stable air stratification allows only weak circulation in the deep cavity (WS/1) despite occasionally strong winds. Rapid surface cooling destabilizes the air column and produces buoyancy-driven circulation (Rayleigh ventilation ④). Vigorous mechanical mixing of the isothermal, labile air column by strong winds rarely occurs because it requires an isothermal air column under snow free conditions. **(c)** Temperatures. (The circled numbers ① and ④ refer to Table 3).

In winter (Fig. E1; not shown in Fig. 11), the amount of snow controls the strength of the air circulation and possibly also the pathways. The thicker the snow cover and the stronger the decoupling between AL and atmosphere (AL–atmosphere coupling in Amschwand et al. (2024a)), the more important density contrasts become to drive the air circulation (buoyancy-driven ventilation), however at overall lower airflow speeds (Fig. E2b–c). Under a thick snow cover in winter 2020–2021, AL

airflow is weak and beneath the level of detection at all sensors and depths. For example, the wind hole instrumented with the WS/5 was completely snowed up and closed. In the snow-poor winter 2021–2022, AL circulation resumed in December 2021 one month after the onset of the snow cover. Measured air circulation is most vigorous and persistent in a rock-glacier furrow, a topographic depression (WS/6 in Fig. E1), and tends to increase with depth. The airflow follows a regular diurnal cycle with nocturnal speed peaks and calm days, in phase with thermally driven (katabatic) local slope winds (Amschwand et al., 2024a). The timing and vertical speed profile of winter-time diurnal oscillation is opposite to the 'summer mode', however much weaker and not as regular as in summer.

## 5.6 Seasonal AL ice turnover

### 5.6.1 Stake measurements

The ground ice is rarely accessible in coarse blocky landforms. Here, we present one of few (to our knowledge) in-situ measurements of the seasonal turnover of superimposed AL ice in rock glaciers and periglacial landforms like block fields (Sawada et al., 2003; Marchenko et al., 2012, 2024)). The ground-ice table as measured at the stake in a rock-glacier furrow deepened by $60\,\mathrm{cm}$ during the thaw season Jun–Sep 2022, rose by (at least) $40\,\mathrm{cm}$ in winter 2022–2023, deepened again by $40\,\mathrm{cm}$ in Jul–Sep 2023, and rose by $60\,\mathrm{cm}$ in winter 2023–2024 (Fig. 12a–c). The stake measurements show no local AL thickening for the years 2022–2024. At least locally in the rock glacier furrow, the ground ice that melted during each thaw season was regenerated by trapping in-blown snow and refreezing snowmelt in the following winter and spring, resulting in a substantial turnover (build-up and melt) of $\Delta\zeta \approx \sim 60\,\mathrm{cm}$ within the coarse blocky AL (equivalent to $\Delta\zeta\,\phi_{al}\,\rho_i = 220\,\mathrm{mm}$ water equivalent w.e.). Note that the lowering of the ground ice table is observed within the blocky AL, i.e. needs to be multiplied by the AL porosity $\phi_{al}$ and ice density $\rho_i$ to obtain an ablation in the glaciological sense (in water equivalent).

The amount of ice lost due to melt is equivalent to a heat flux $\bar{Q}_m$ of $\sim 10\,\mathrm{W\,m^{-2}}$ on average during the 2022 thaw season (porosity $f_i = \phi_{al} = 0.4$ in Eq. 7, Fig. 12d), in good agreement with $Q_{\mathrm{CGR3}}^{rad}$ (Fig. 8). The melt/thaw rates accelerate and decelerate with a peak in mid-July, proportional to the ground surface temperature $\mathrm{d}\zeta/\mathrm{d}t := \dot{\zeta} \propto T_s$ (within their uncertainty) throughout the thaw season (Fig. 12d). Measured melt rates are independent of the time elapsed since onset of the thaw season. This justifies the two-layer Stefan equation (Eq. 12), because the one-layer Stefan equation (Eq. 10) predicts $\dot{\zeta} \propto \sqrt{t}$, with thaw rates rapidly slowing down as the thaw front recedes away from the surface.

### 5.6.2 Stefan parameterisation

With $k_{\mathrm{eff}} = 3.0 \pm 0.3\,\mathrm{W\,m^{-1}\,K^{-1}}$ a priori derived from our measurements (Sect. 5.4) and $f_2 = \phi_{al} = 0.4$ (saturation), the best-fit parameters for Eq. 12 are $\hat{f}_1 = 0.01$ and $\hat{h}_1 = 3.0 \pm 0.25\,\mathrm{m}$ (Fig. 12a). This relation based on 2022 data predicts the 2023 ablation rates well (Fig. 12b). The estimated $\hat{h}_1$ is $50\%$ thicker than the actual distance to the initial ground-ice table ($\sim 2\,\mathrm{m}$), but still plausible given the rough terrain and input data uncertainties: The "excess" overburden/insulation might compensate for the likely too high forcing thaw index $I_t(T_s)$ in the shaded furrow, as $I_t(T_s)$ is derived from the PERMOS outgoing long-wave radiation on a sun-exposed plateau.

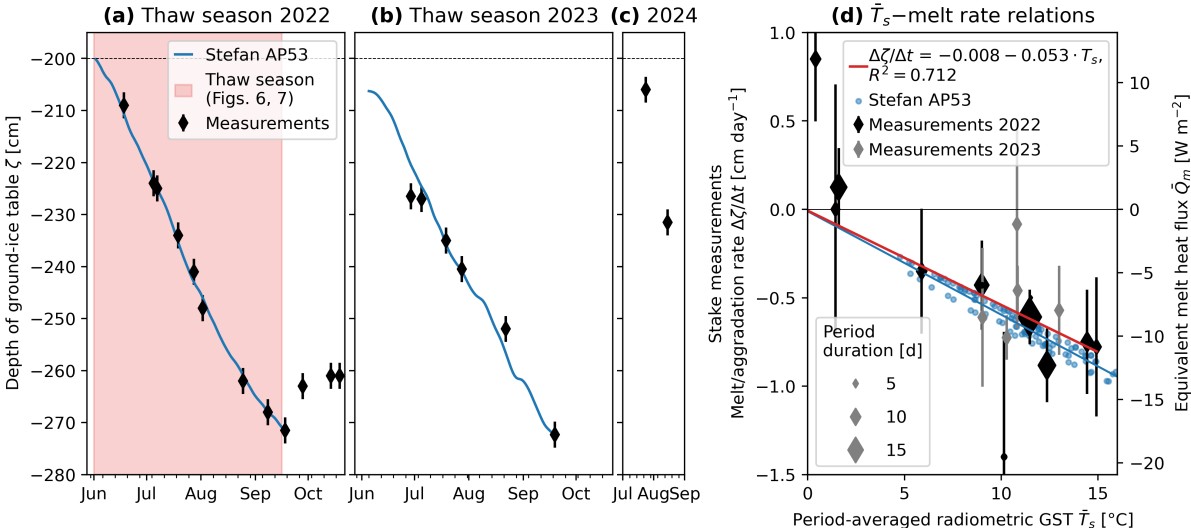

**Figure 12.** Observed vertical changes in the ground-ice table in thaw season **(a)–(c)** 2022–2024 with seasonal build-up and melt. Measurement uncertainty $\sim 5$ cm. Melt is simulated with the Stefan model (Eq. 12, Fig. 4). **(d)** The ground-ice melt rates are correlated with the ground surface temperature $T_s$.

## 6 Discussion

### 6.1 AL energy budget: Thaw-season heat uptake and partitioning

#### 6.1.1 Heat uptake driven by earlier snow melt-out and hot–dry summer weather

During the thaw season, the AL is a net heat sink. Fig. 14 shows the cumulative heat uptake during the two thaw seasons 2021 and 2022. At the onset of the thaw season which coincides with the disappearance of the snow cover, the AL exits the zero-curtain phase isothermal at $0°C$ (Fig. 7a). This is the thermodynamic zero level $0$ MJ m$^{-2}$ in Fig. 14. The sensible heat content $H_{al}^{\theta}$ is zero at the onset and end of the thaw season. During the thaw season, the AL is a heat sink that absorbs roughly 10% of the surface net radiation $Q^*$ (not to be confused with the below-ground long-wave/thermal net radiation $Q_{\mathrm{CGR3}}^{rad}$; Fig. 14), hence $Q_G \approx Q^*/10$ (Amschwand et al., 2024a), and that in both thaw seasons until mid/end August. Roughly $90\%$ of $Q^*$ is exported back into the atmosphere via sensible and latent turbulent fluxes (Fig. 13 in Amschwand et al., 2024a), but the rock glacier is vulnerable to hot–dry weather spells.

The total heat uptake during the thaw season is first controlled by the date of the thaw season onset, that is, the snow melt-out date, in turn controlled by winter precipitation and spring weather. Also a warm autumn extends the thaw season and September heat waves can bring in heat almost at August rates (Aug–Sep 2023, Table 4), but the impact is presumably less severe than an earlier onset in spring because solar radiation is then less intense. After the snow-poor winter 2021–2022, the thaw season 2022 started one month earlier than in 2021 and received almost twice the amount of heat, $93.7$ MJ m$^{-2}$ instead of $52.1$ MJ m$^{-2}$,

although the thaw season lasted only 15 days longer (Table 5). A one month earlier snow melt-out, beginning of June instead of beginning of July, caused a heat uptake of $\sim 40\,\mathrm{MJ\,m^{-2}}$, amounting to $\sim 40\,\%$ of the entire 2022 heat uptake. Alone a snow-free June brings in over $70\,\%$ of the entire 2021 heat uptake. Weather conditions during the thaw season also matter:
Hot–dry weather spells deplete the water and ice stores in the AL, decrease its latent buffer capacity (Sect. 6.2), and enhance the downward heat transfer (heat uptake ratio by $Q_{\mathrm{CGR3}}^{rad}$, Fig. 14b2) (Amschwand et al., 2024a). Based on $Q_{\mathrm{CGR3}}^{rad}$, the July 1– Sep 15 heat uptake (period common to both thaw season) was in 2022 $16.8\,\mathrm{MJ\,m^{-2}}$ higher than in the $2.3\,^\circ\mathrm{C}$ cooler summer 2021 (Table 4), $33\,\%$ more, and roughly corresponding to ten June days worth of heat uptake. Importantly, also proportionally more heat from the surface net radiation $Q^*$ was transferred to deeper AL levels, $8.9\,\%$ (of $748\,\mathrm{MJ\,m^{-2}}$) in 2022 compared
to $7.5\,\%$ (of $664\,\mathrm{MJ\,m^{-2}}$) in 2021. Thaw season 2023 was overall similar to 2022, except for a warmer September.

Hence, two forcings mainly control the total heat uptake of the AL during the thaw season, (i) its date of onset, and (ii) weather conditions. With our quantitative data, we can attempt to estimate how strongly each of these forcings impacts Murtèl by unravelling how much heat is taken up by the AL in response to (i) the trend towards earlier melt-out and (ii) the warming trend (Table 4). First, the June heat uptake scales with $0.9\times$ the number of snow-free June days (Fig. 13b), i.e. has
535 a sensitivity of $1.1\,\mathrm{MJ\,m^{-2}}$ per snow-free June day. Second, the July 1–Sep 15 heat uptake scales with $6.6\times$ the air temperature increase with respect to the 2021 average $\bar{T}_a$ (Fig. 13c), i.e., a sensitivity of $6.6\,\mathrm{MJ\,m^{-2}}$ per $^\circ\mathrm{C}$ of summer warming. Translating these sensitivities to trends should be interpreted with utmost caution because the snow melt (Matiu et al., 2021) and warming trends have a spatio-temporal variability, have accelerated in the recent decades, and the climate sensitivity is itself sensitive to the evolving AL properties (e.g., negative feedback by AL thickening (Haeberli et al., 2024), altered SEB).
First, Hoelzle et al. (2022) (for Murtèl 1997–2018, Fig. 13a), Klein et al. (2016), and Matiu et al. (2021) report trends of earlier snowmelt of $1-5\,\mathrm{days\,decade^{-1}} \times 1.1\,\mathrm{MJ\,m^{-2}\,day^{-1}} = 1.1-5.5\,\mathrm{MJ\,m^{-2}\,decade^{-1}}$. Second, a warming trend (Hoelzle et al., 2022) of $0.4-0.7\,^\circ\mathrm{C\,decade^{-1}} \times 6.6\,\mathrm{MJ\,m^{-2}\,(^\circ C)^{-1}} = 2.6-4.6\,\mathrm{MJ\,m^{-2}\,decade^{-1}}$. The total AL heat uptake has likely been increasing by $4-10\,\mathrm{MJ\,m^{-2}}$ per decade ($4-11\,\%$ of the 2022 heat uptake of $94\,\mathrm{MJ\,m^{-2}}$). This calculation is (to our knowledge) the first quantitative attempt to express the climate sensitivity of a rock glacier in numbers based on in-situ
heat-flux measurements instead of modelling.

Above calculation refers only to the thaw season heat uptake. However, to fully assess the impact of climate change on Murtèl rock glacier (Scherler et al., 2013), the winter cooling needs to be accounted for as well, and that both for the AL and the permafrost body beneath. In a coarse blocky AL, only the amount of cold content that is converted to ground ice contributes to offsetting the heat uptake during the thaw season. No sensible cold content ($T_{al} < 0\,^\circ\mathrm{C}$) can be retained in
the highly permeable AL flushed by snowmelt and warmed to $0\,^\circ\mathrm{C}$ in spring (isothermal entering the zero curtain, Fig. 7). AL ice build-up is discussed in Sect. 6.2. The second process is building cold content of the permafrost body (rock glacier core) beneath by (preferentially convective) heat export through the AL and the snowpack (Luetschg et al., 2008). We discuss winter-time heat transfer in Sect. 6.4, while the year-round energy budget of the entire rock glacier (AL and permafrost body) is beyond the scope of this study.

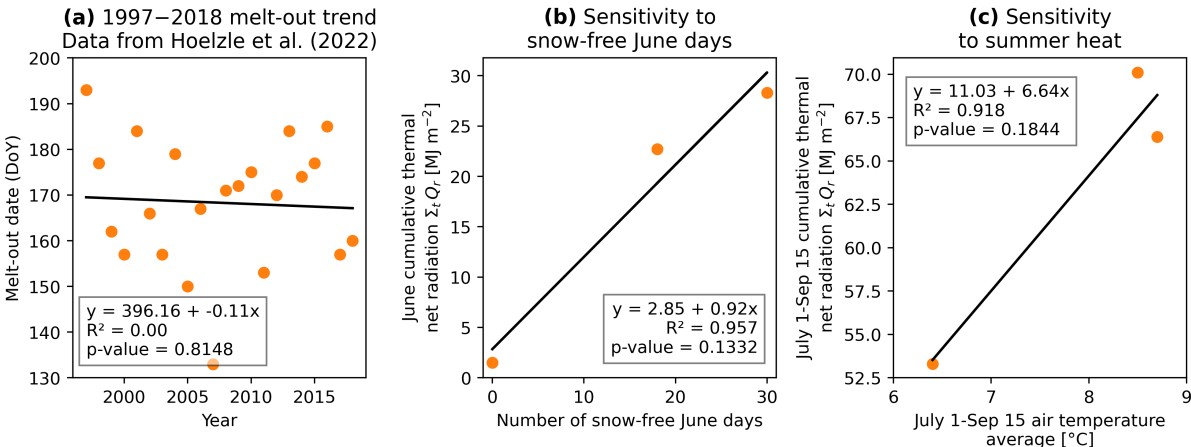

**Figure 13. (a)** Melt-out trend at Murtèl (data from Hoelzle et al. (2022)). The trend of $-0.1 \text{ days yr}^{-1}$ is not statistically significant because the time series is short compared to the inter-annual variability, but is likely to continue with ongoing climate change. **(a–b)** 2021–2023 sensitivity of the AL heat uptake: **(b)** The more snow-free days in June and **(c)** the warmer the Jul–Sep period, the larger is the heat uptake.

**Table 4.** Monthly average air temperatures $\bar{T}_a$ [°C], cumulative heat uptake $\Sigma_t Q_r$ [MJ m$^{-2}$], and average daily heat uptake rate $\Sigma_t Q_r / \Delta t = Q_r$ [MJ m$^{-2}$ d$^{-1}$ = 11.57 W m$^{-2}$] for the thaw seasons 2021–2023. Heat uptake into the AL is most intense in July.

| | Thaw season 2021 | | | Thaw season 2022 | | | Thaw season 2023 | | |
|---|---|---|---|---|---|---|---|---|---|
| | $\bar{T}_a$ | $\Sigma_t Q_r$ | $Q_r$ | $\bar{T}_a$ | $\Sigma_t Q_r$ | $Q_r$ | $\bar{T}_a$ | $\Sigma_t Q_r$ | $Q_r$ |
| Thaw season[a] | 5.9 | 54.6 | 0.6 | 8.0 | 94.4 | 0.9 | 8.0 | 89.8 | 0.9 |
| July 1–Sep 15 | 6.4 | 53.3 | 0.7 | 8.5 | 70.1 | 1.0 | 8.7 | 66.4 | 0.9 |
| June | 5.7 | 1.5 | 0.1 | 7.7 | 28.3 | 0.9 | 6.4 | 22.7 | 0.8 |
| July | 7.1 | 24.6 | 0.8 | 9.7 | 36.8 | 1.2 | 8.6 | 29.6 | 1.0 |
| August | 6.2 | 20.1 | 0.6 | 8.5 | 27.0 | 0.9 | 8.3 | 23.9 | 0.8 |
| September[b] | 5.2 | 10.3 | 0.5 | 4.1 | 7.2 | 0.4 | 6.7 | 14.3 | 0.7 |
| October[a] | −1.0 | 1.3 | 0.0 | 2.8 | 6.6 | 0.2 | | | |

[a] In 2023: data until Sep 20 (rock fall). [b] Period Sep 1–20, limited by 2023 data.

### 6.1.2 Heat partitioning

The available heat from the surface ground heat flux $Q_G$ is partitioned into sensible heat storage changes $\Delta H_{al}^{\theta}$, latent heat storage changes (ice melt) $Q_m$, and conducted into the permafrost body beneath the AL $Q_{\text{PF}}$ (Eq. 4, Table 5). On thaw-season average, $Q_G$ is largely ($\sim 70\%$) spent on melting ground ice. Hence, *latent heat effects contribute substantially to the thermal buffering — this is one process that renders rock glaciers climate-robust*, as long as seasonal build-up of superimposed ice

compensates for its melt (discussed in Sect. 6.2). Roughly $\sim 20\%$ of $Q_G$ is absorbed by the coarse blocky AL as sensible heat storage $H_{al}^{\theta}$. The heat conducted into the permafrost body beneath the AL $Q_{\mathrm{PF}}$ amounts to $\sim 10$ MJ m$^{-2}$ ($\sim 10\%$ of $Q_G$), about $1\%$ of the available net radiation $Q^*$ at the surface ($Q_{\mathrm{PF}}/Q^*$ in Table 5). The cumulative rain heat flux $Q_{Pr}$ (Eq. 5) is 11 MJ m$^{-2}$ in the cool-wet summer 2021. $Q_{Pr}$ is a small flux compared to $Q_G$ in 2022 ($5-10\%$ considering the rainfall undercatch), but not in 2021 ($20\%$), and is in any case similar to $Q_{\mathrm{PF}}$. Hence, the rain heat flux $Q_{Pr}$ has a weak cooling effect near the surface ($Q_{Pr} < Q_m, Q_G$), but potentially an important warming effect at depth ($Q_{Pr} \approx Q_{\mathrm{PF}}$).

**Table 5.** Thaw-season average (avg) and cumulative total (cum) heat partitioning.

|  | Thaw season 2021 | | Thaw season 2022 | |
| --- | --- | --- | --- | --- |
| duration | 95 days | | 110 days | |
| [MJ m$^{-2}$] | avg | cum | avg | cum |
| $Q^*$ (SEB) | 444 | 753 | 633 | 1136 |
| $Q_G$ (SEB) | 37.3 | 52.1 | 64.9 | 93.7 |
| $Q_r$[a] | 32.0 | 54.6 | 51.6 | 94.4 |
| $H_{al}^{\theta}$ | 9.5 | 0.0 | 20.7 | 0.0 |
| $Q_{\mathrm{PF}}$ | 4.0 | 7.4 | 7.3 | 12.7 |
| $Q_{Pr}$ | 6.9 | 11.0 | 1.5 | 4.7 |
| dev$_{al}$[b] | 30.7 | 55.7 | 38.4 | 85.7 |
| $Q_m$ (AP53)[c] | 31.4 | 55.9 | 52.7 | 88.5 |
| *Ratios* | | | | |
| $Q_m/Q_G$ | 0.84 | 1.07 | 0.81 | 0.94 |
| $Q_G/Q^*$ | 0.08 | 0.07 | 0.10 | 0.08 |
| $Q_{\mathrm{PF}}/Q^*$ | 0.01 | 0.01 | 0.01 | 0.01 |

[a] $Q_r$ is the pyrgeometer measurement $Q_{\mathrm{CGR3}}^{rad}$. [b] AL energy budget deviation dev$_{al} := Q_G + Q_{Pr} - H_{al}^{\theta} - Q_{\mathrm{PF}}$. [c] Ablation parameterised via Eq. 12 and converted to $Q_m$ via Eq. 7. The 2021 ablation where no observation are available is estimated with the 2022 parameters and the 2021 forcing $T_s$.

The thaw season is divided in two phases, an AL heating and a ground ice melting phase (Fig. 14a2, b2). Initially, the ice-poor shallow AL is heated from the surface downwards. During the first phase, the uptake of sensible heat takes $2-3$ weeks to saturate at an average $H_{al}^{\theta}$ after which the sensible heat storage changes little (even slowly loses heat in late summer), and the heat goes mainly into ice melt. Due to the shallow AL and steep temperature gradients shortly after the thaw season onset, $Q_{\mathrm{CGR3}}^{rad}$ is relatively large compared to $\Delta H_{al}^{\theta}$ (Fig. 14a2, b2). In the second phase, the near-surface AL does still warm and cool in response to the atmospheric forcing, but the sensible heat storage changes $\Delta H_{al}^{\theta}$ are small compared to the cumulative heat uptake.

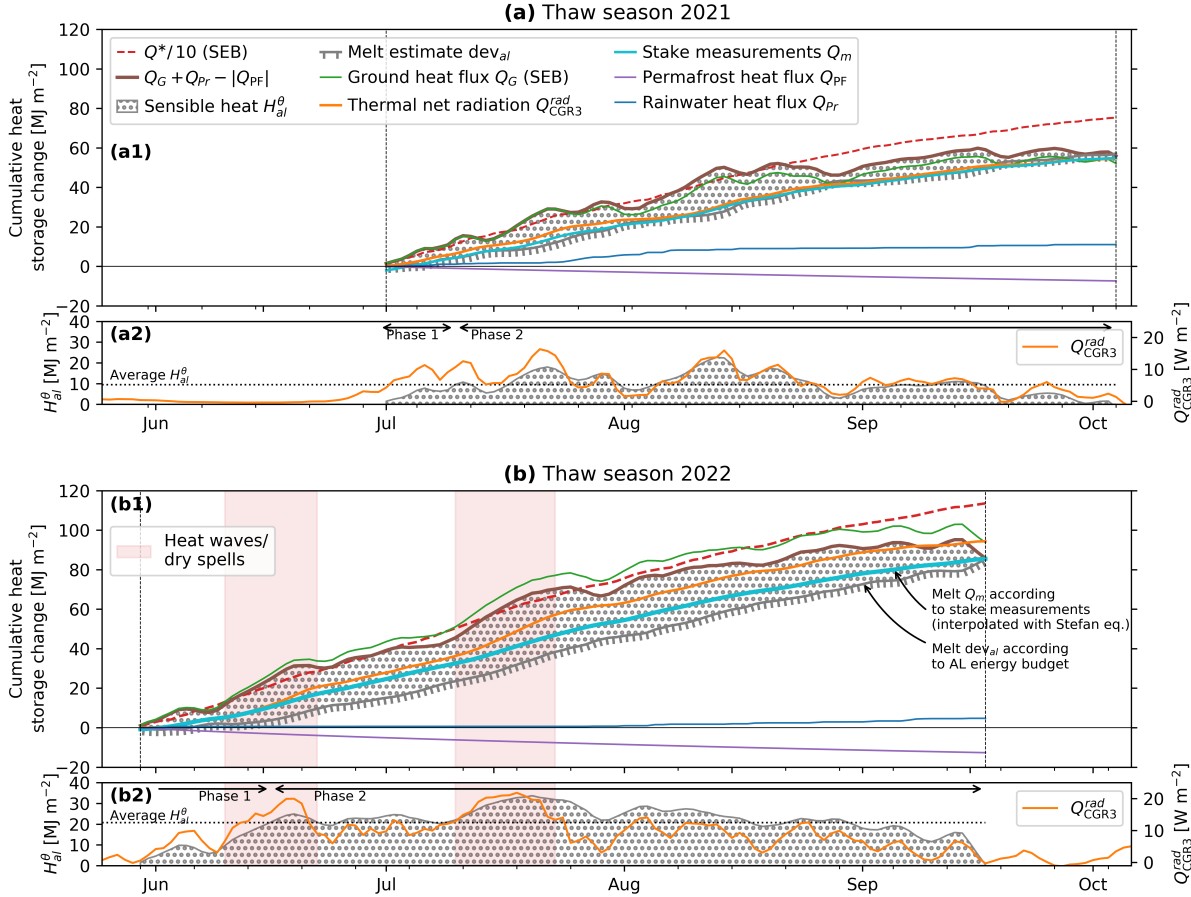

**Figure 14.** Heat uptake and partitioning during the **(a)** 2021 and **(b)** 2022 thaw seasons. Most of the heat supplied to the AL is intercepted by melting ground ice. SEB refers to the Amschwand et al. (2024a) surface energy balance. **(a2, b2)** Sensible heat storage $H_{al}^{\theta}$ and thermal net radiation $Q_{\text{CGR3}}^{rad}$ are correlated. Phases 1 and 2 are explained in the text. The marked 2022 heat waves are also shown in Fig. 5.

## 6.2 The seasonal ice turnover in the AL: ice protects the underlying permafrost

The 2022 stake measurements (Fig. 12a–c) in the rock glacier furrow (interpolated and converted to $Q_m$ using Eqs. 12 and 7, respectively) agree within $10\ \text{MJ}\ \text{m}^{-2}$ with the melt $\text{dev}_{al}$ calculated from the AL energy budget in the nearby instrumented cavity (Eq. 4; Fig. 14b, Table 5). The AL budget, estimated in a broad ridge with thicker AL, predicted more sensible storage $H_{al}^{\theta}$ gains and less ice melt $\text{dev}_{al}$ than the stake measurements $Q_m$ show for the narrow rock-glacier furrow (cf. Fig. 2). The discrepancy relative to the total heat uptake decreases during the thaw season, end-of-thaw season estimates match. Although our plot-scale observations do point at some differential melt beneath furrows and ridges (a micro-topographic variability mentioned by Kääb et al. (1998) and Halla et al. (2021)), the agreement suggests that our estimates of end-of-thaw season ice storage changes are fairly representative over the landform within an uncertainty that we estimate as $\pm 30\ \%$. No system-

atic stake measurements were taken in 2021, $Q_m$ is estimated with the 2022 parameters forced by the 2021 $T_s$ (Fig. 14a). Differential heat storage effects are smaller in the cooler thaw season 2021.

The AL ice was fully regenerated and no *net* ice loss occurred between Sep 2020–Sep 2024 at least locally in the furrow. While this is not exactly true for the entire Murtèl rock glacier where slow permafrost degradation and AL has been measured (Noetzli and Pellet, 2023), Murtèl's slow response testifies the important latent buffer effect of the AL ice (Sect. 6.1), a feature common to ice-rich permafrost landforms (Scherler et al., 2013). The regeneration of ground ice in the AL partly explains the climate robustness of coarse blocky landforms (Scherler et al., 2013). If the lost ground ice is not regenerated, the permafrost landform is preconditioned towards AL thickening and irreversible degradation (Hilbich et al., 2008; Hauck and Hilbich, 2024). Moreover, the modelling study by Renette et al. (2023) suggests that dry cooling in early winter and ice build-up in spring, a timing specific to permeable and well-drained (sloped) permafrost landforms, is itself an undercooling process, additional to convective undercooling (Sect. 6.3.1). This dry undercooling effect is most pronounced in deeply snow-covered landforms where the autumn–early winter "window of opportunity" for cooling before the onset of an insulating snow cover is shorter. The intricate relations between hydraulic and ground thermal regimes in coarse blocky permafrost landforms and the AL ice as a meltwater "source" in hot–dry summer periods are discussed further in Amschwand et al. (2024b).

## 6.3 Thaw-season heat transfer

### 6.3.1 The thermal semi-conductor effect: Air convection selectively enhances the apparent thermal diffusivity during cooling events

The sensitivity of the apparent thermal diffusivity $\kappa_a$ to AL temperature gradients reflects how efficient convective heat transfer operates compared to radiation–conduction in the coarse blocky AL. $\kappa_a$ is primarily controlled by the AL air column stability (vertical temperature gradient $\nabla_z T_{al}$) that induces *buoyancy-driven convection*, and secondarily by the atmospheric wind speed $u$ that induces *wind-forced convection* (Fig. 10) (Herz, 2006). Hence, the convection-enhanced apparent $\kappa_a$ is as much determined by the time-variable meteorological conditions as by the debris texture and thus variable in time. Note that in such permeable material, water content does not affect heat transfer properties. $\kappa_a$ is higher for cooling (upwards heat transfer) at unstable temperature gradients than for warming (downwards heat transfer) at stable temperature gradients. This feedback between AL temperature gradient and thermal diffusivity profoundly impacts the ground thermal regime of permeable, ventilated landforms: frequent, but less efficient radiative-conductive warming (suppressed convection) is countered by only occasionally occurring, but highly efficient convective cooling (enhanced convection; Figs. 8, 10). Ventilation leads to locally lower ground temperatures in coarse blocky, permeable terrain, an observation known as *undercooling* (Wakonigg, 1996), and is another process that renders rock glaciers climate-robust. This effect has long been qualitatively known as 'Balch ventilation' (Balch, 1900) or the 'thermal semi-conductor effect' (Guodong et al., 2007), (cf. Johansen, 1975; Herz, 2006). Our study is the first one (to our knowledge) that quantifies the effect based on field data and calculates a convection-enhanced apparent thermal diffusivity $\kappa_a$. Table 3 provides an overview on how the seasonally varying dominant heat transfer processes are shown by our

data. The impact of air convection is visible in the temperature, airflow speed, and heat flux plate measurements at sub-diurnal

resolution (Appendix B).

Our $\kappa_a$ value agree with published values for ventilated coarse blocky material, but are generally $2-6$ times higher than for finer material of supra-glacial debris (Rowan et al., 2021) or cryic regosol (Appendix Table F1). The important contribution of forced air convection to the total heat transfer even at stable air stratification is characteristic for highly permeable and dry materials, i.e. is specific to coarse blocky landforms, and there most pronounced in the strongly ventilated, wind-exposed

near-surface layer (Yoshikawa et al., 2023). With smaller grain size or increasing fine-material content that clogs the pore space (typically near the AL base), convective and radiative heat transfer (Sect. 6.3.2) becomes less important in favour of conductive heat transfer. We estimate the key parameter intrinsic permeability $K$ using the Kozeny–Carman relation in the Appendix Sect. D.

### 6.3.2 Radiative heat transfer and stagnant effective thermal diffusivity $\kappa_a^0$

Above a temperature gradient of $4 \ \mathrm{K \ m^{-1}}$, turbulence is suppressed to the point where the effective thermal diffusivity $\kappa_a$ becomes independent of $\nabla_z T_{al}$ (Fig. 10). The thermal stratification inside the AL becomes too stable to be mixed by the wind and wind-forced convection is "switched off" at a temperature gradient threshold, slowing down an "overheating" of the AL. This 'non-linear heating of the AL with air temperature' has been reported by Hanson and Hoelzle (2004) and Herz (2006). Our threshold temperature of $8-10°\mathrm{C}$ (ca. 2 m above the AL base at $0°\mathrm{C}$) is higher than the $6°\mathrm{C}$ threshold reported by Hanson and

Hoelzle (2004) for a less coarse blocky measurement spot on Murtèl. Perhaps the higher permeability around our instrumented cavity imposed less resistance to wind-forced mixing for a given temperature gradient.

Moreover, this $\kappa_a$ under strongly stable air stratification is our best-available field estimate of the *stagnant* effective thermal diffusivity $\kappa_a^0 = 9.6 \times 10^{-7} \ \mathrm{m^2 \ s^{-1}}$ ($k_{\mathrm{eff}}^0 \approx 1.2 \ \mathrm{W \ m^{-1} \ K^{-1}}$), i.e. a radiative–conductive thermal diffusivity without convection. This $k_{\mathrm{eff}}^0$ is $\sim 3\times$ higher than what would be expected from the geometric mean or empirical engineering parameterisations

that ignore radiation, for example Johansen (1975)'s $k_{dry} = 0.039 \, \phi^{-2.2} \pm 25\%$ for dry crushed rock (Côté and Konrad, 2005). A relatively large $\kappa_a$ uncertainty of $\sim 30\%$ (mainly due to variable block sizes) does not detract from this finding. Hence, an important insight for modellers is that the stagnant thermal diffusivity $\kappa_a^0$ of coarse blocky material is underestimated if radiative heat transfer is ignored. Our measurements confirm previous investigations on Murtèl in that respect (Scherler et al., 2014; Schneider, 2014) and is further supported by the cold-region engineering study by Fillion et al. (2011). The radiative

thermal diffusivity $\kappa_a^0$ increases linearly with block/pore size (actually: the effective length for radiation in the air-filled gaps between particles) and mean temperature cubed (Lebeau and Konrad, 2016), i.e., tends to counteract undercooling (quantitative details in Appendix Sect. D). Laboratory tests using crushed rock beds showed that radiative heat transfer begins to dominate over conduction at effective particle sizes ($d_{10}$ diameter) exceeding $9 \ \mathrm{cm}$ (cobbles) (Fillion et al., 2011; Rieksts et al., 2019), corroborating an earlier work by Johansen (1975).

### 6.4 Autumn and winter-time heat transfer

The early-winter snow cover determines the ground thermal regime in winter and spring by controlling the magnitude of the heat fluxes and convective air exchange across the snow cover via *snow funnels*. In terms of qualitative process understanding, this is established knowledge (Haeberli et al., 2006; Wagner et al., 2019) and is shown on Murtèl by the permafrost temperature time series since 1987 (Noetzli and Pellet, 2023): Strong ground cooling during snow-poor winters can offset the warming of the preceding years in the permafrost body to more than $20\,\mathrm{m}$ depth. The degree of snow-cover insulation is shown in our data by the two contrasting winters 2020–2021 (average snow conditions, weak air circulation beneath a closed snow cover) and 2021–2022 (snow-poor winter, strong air circulation beneath a semi-closed snow cover). Although the air column in the somewhat insulated AL and in a thin, strongly cooled layer above the snow surface was typically nonlocally unstable (i.e., near-surface $T_a \approx T_s < \max\{T_{al}\}$, Fig. 7a) and the potential for buoyancy-driven convection was available, different air circulation patterns emerged depending on snow height: (i) *Rayleigh ventilation* (Marchenko, 2001; Millar et al., 2014) prevailed in unresisted circulation, (ii) *cold-air infiltration* (Herz, 2006) occurred beneath a moderately thick/semi-closed snow cover, and (iii) stagnant–conductive conditions without air circulation occurred beneath a thick/closed snow cover. The circulation patterns differ in terms of persistence in time, heat flux magnitude, vertical temperature profile, and Rayleigh numbers (local instability). Hence in addition to the temperature profile, the AL–atmosphere connectivity through the snow cover ('effective aeraulic resistance') co-controlled which type of air circulation occurred, and ultimately how strong the winter cooling was.

#### 6.4.1 Rayleigh circulation under snow-free conditions or beneath a thin/open snow cover

*Rayleigh ventilation* events occured typically in autumn before the onset of a thick snow cover, for example in Oct 2020 (Fig. 6b, Table 3④). With unresisted AL–atmosphere exchange, it is an efficient ($20-30\,\mathrm{W\,m^{-2}}$, Fig. 8) top-down cooling process associated with the characteristic negative AL temperature gradients (locally unstable air stratification, Fig. 7a, Table 3④) and is diagnosed by supercritical Rayleigh numbers ($\mathrm{Ra} > \mathrm{Ra}_c$). Rayleigh ventilation events as a response to rapid atmospheric cooling are a short-lived, but efficient heat transfer process. Thermal equilibrium was reached rapidly within hours–days, for example in Sep 2020 or 2022. It contributed to the rapid end of the 2022 thaw season, where the entire AL was cooled from 5 to $0°C$ within one day.

#### 6.4.2 Cold-air infiltration beneath a semi-closed snow cover

During extended *cold-air infiltration* phases with a semi-closed (patchy) snow cover ($h_S < 60\,\mathrm{cm}$, Fig. 5b; Amschwand et al. (2024a)), the AL cooled bottom-up slowly and persistently over longer periods (days–weeks) at moderate fluxes ($\leq 10\,\mathrm{W\,m^{-2}}$, $Q_{\mathrm{CGR3}}^{rad} \not\propto Q_{\mathrm{HFP}}$, Fig. 8, Table 3③). Cold-air infiltration shaped the ground thermal regime in November 2020 and throughout the snow-poor winter 2021–2022. It caused $5°C$ lower AL temperature minima compared to winter 2020–2021, although winter 2021–2022 was $0.4°C$ warmer (Nov-Mar average). Convective exchange with the atmosphere is shown by fluctuating AL temperatures and characteristic concave temperature profiles with a minimum at mid-cavity level ('bulges', Fig. 7a–b, Table 3③, Herz et al. (2003b)) whose depth coincides with increased daily temperature amplitudes (Fig. 7b). Cooling at depth

stabilized the AL air column, shown as subcritical Rayleigh numbers (Ra < Ra$_c$), and lead to a net downward radiative transfer $Q_{\mathrm{CGR3}}^{rad} > 0$ like during the thaw season (although much smaller), opposite to the measured HFP/1 heat flux $Q_{\mathrm{HFP}}$ (Fig. 8). Modelling convective heat exchange with the Rayleigh number alone would miss this type of air circulation. The bottom-
680 up cooling was accompanied by a bottom-up drying, since ventilation brought in 'fresh', dry outside air into the otherwise saturated AL (Fig. 15, Fig. 7c–d ③), opposite to the summertime evaporative top-down drying. In-phase diurnal oscillations of AL relative humidity, temperature differences between AL and surface temperatures ($T_s - \min\{T_{al}\}$), and strong nighttime ventilation recorded in the rock-glacier furrow (WS/6 in a topographic depression, Fig. E1) suggest that cold-air infiltration occured in clear-sky nights. Radiatively cooled air on the snow surface, produced by the nocturnal negative radiation balance
(Amschwand et al., 2024a), infiltrated into the coarse blocky AL (Herz, 2006). Cold-air infiltration is an effect of nonlocal static instability (Stull, 1991) that arises from interactions between AL and a semi-closed snow cover. The process is analogous to the summertime nocturnal near-surface air circulation that switches on when the near-surface atmosphere cools below the near-surface AL (nocturnal Balch ventilation, Amschwand et al. (2024a)). Although our isolated point-wise measurements could not reveal the lateral extent and connectivity of the air flow and we did not perform gas tracer tests (Popescu et al., 2017a),
the cold-air infiltration likely corresponds to the landform-scale *cold-air drainage* described in the literature (Wakonigg, 1996; Delaloye and Lambiel, 2005; Millar et al., 2014) where the infiltrating cold air flows laterally downslope in the permeable AL beneath the snow cover (convection–advection), analogous to the katabatic drainage flows on the snow cover (Amschwand et al., 2024a).

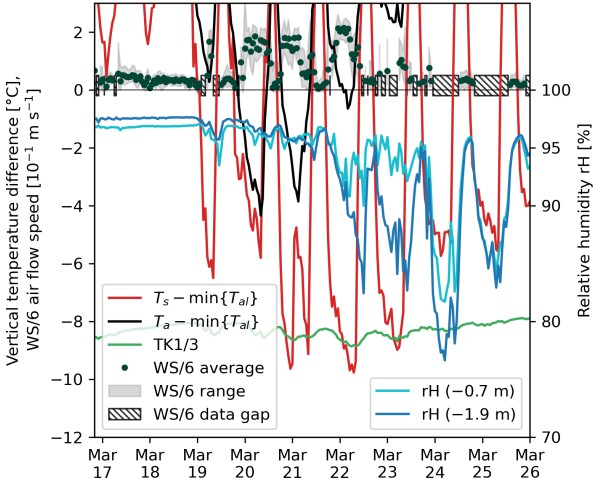

**Figure 15.** Nocturnal cold-air infiltration episodes in March 2022 as indicated by airflow speed measurement (WS/6) and a simultaneous drop of AL relative humidity (HV5/1–2) and temperature (TK1/3) due to the ventilation with fresh, dry-cold outside air. As soon as the ventilation stops, the AL air approaches saturation within hours. Higher WS/6 airflow speeds always coincide with negative ($T_s - \min\{T_{al}\}$). Using air temperature $T_a$ instead of snow surface temperature $T_s$ would underestimate the occurrence of cold-air infiltration episodes. Note the WS/6 data gaps due to power shortage.

### 6.4.3 Stagnant conduction beneath a thick/closed snow cover

In the more snow-rich winter 2020–2021, after closing of the snow cover in December ($h_S > 60$ cm, Fig. 5b), heat fluxes were small ($< 2$ W m$^{-2}$) and upwards (Fig. 8, Table 3②; 'closed/insulating snow cover' sensu Amschwand et al. (2024a)). Heat transfer on a daily timescale appeared diffusive ($Q_{\mathrm{CGR3}}^{rad} \propto Q_{\mathrm{HFP}}$). The AL heat flux was not larger than the conductive heat flux $Q_S$ across the snow cover as calculated in Amschwand et al. (2024a). The AL air column was near-isothermal (Fig. 7a, Table 3③) and weakly unstable (subcritical Rayleigh numbers in Fig. 6b). The measurements of heat fluxes and airflow speed were close to their instrumental accuracy.

### 6.5 Scope and transferability of Murtèl findings

The findings of this detailed single-site case study are transferable to other sites to varying degrees, which is a key consideration for upscaling. Caution is necessary because convective and radiative heat transfer and the ice build-up are specific to a high permeability, large pore dimensions, and dry conditions, which are in turn characteristic of a coarse blocky debris texture. First, most transferable in the sense that it is the least sensitive to the debris texture is the SEB and the AL energy budget (i.e., the *total* heat uptake) that more strongly reflects topo-climatic and snow conditions (Amschwand et al., 2024a). The effects of earlier spring melt-out and warmer summers, exacerbated by dry periods, strongly impact the AL energy budget primarily via the surface net radiation and sensible turbulent fluxes (Amschwand et al., 2024a). Moisture storage, transfer, and evaporation (latent turbulent fluxes) are more sensitive to debris texture. Second, the two undercooling processes – heat interception by the AL ice turnover and convective thermal semi-conductor effect – are characteristic features of all coarse blocky permafrost landforms and shape the ground thermal regime in different debris permafrost landforms including rock glaciers, frozen talus slopes, and block fields as long as their AL is permeable and dry. Hence, the detailed process understanding gained on Murtèl qualitatively applies to widespread mountain permafrost landforms located in various topo-climatic conditions and give a process-oriented, field-data based explanation for the climate robustness of undercooled 'cold rocky landforms' (Brighenti et al., 2021). Only the spatial pattern of air circulation differs in sloped landforms such as talus slopes (Caltagirone and Bories, 1985; Guodong et al., 2007): Convective heat transfer is then no longer dominantly vertical as on Murtèl. Surface-parallel (lateral) advective heat transfer leads to an undercooled foot and an 'overwarmed' top of the slope ('chimney effect') (Delaloye and Lambiel, 2005; Morard et al., 2010; Růžička et al., 2012; Wicky, 2022; Zegers et al., 2024). Third, the least transferable are the exact values of the heat transfer parameters ($k_{\mathrm{eff}}, \kappa_a$) which are so sensitive to the debris texture that they typically vary even on the landform itself (Appendix Sect. D). The numbers are only valid for landforms similar to Murtèl in terms of debris texture.

## 7 Conclusions

We investigated heat transfer and storage processes in the ventilated coarse blocky active layer (AL) of the seasonally snow-covered Murtèl rock glacier situated in a cirque in the Upper Engadine (eastern Swiss Alps). In the highly permeable AL,

conductive/diffusive heat transfer including thermal radiation, non-conductive heat transfer by air circulation, and heat storage changes from seasonal build-up and melting of ground ice create a cool–stable ground thermal regime known as *undercooling*, rendering these permafrost landforms comparatively robust against climate change. While the undercooling effects have long been known qualitatively, this study resolves different processes quantitatively, providing insights into the capability and limits of the undercooling effect and on the climate robustness of coarse blocky landforms. We provided estimates of sub-surface heat flux and storage changes for the two-year period 2020–2022 based on a novel in-situ sensor array in the AL and stake measurements of the seasonal progression of the ground-ice table, i.e., ground ice build-up and melt. The measurements included thermistor strings, hygrometer, heat flux plates, and thermal radiation sensors. Airflow speed sensors (thermo-anemometer) distributed in the AL revealed air circulation patterns. We parameterised the seasonal ground ice melt using a modified Stefan equation, whose key parameter, the effective thermal conductivity, was derived from the in-situ measurements.

This study unravels the two thaw-season processes that render Murtèl rock glacier climate-robust, the seasonal ground ice turnover and convective cooling. First, the coarse blocky AL intercepts $\sim 90\%$ of the thaw-season ground heat flux of $\sim 5-15\,\mathrm{W\,m^{-2}}$ by melting ground ice ($\sim 70\%$; latent storage change that leaves the system as meltwater) and by heating the rock mass ($\sim 20\%$; sensible storage change). A smaller fraction ($\sim 10\%$) is transferred into the permafrost body beneath and causes slow permafrost degradation. The cumulative heat uptake of $\sim 50-90\,\mathrm{MJ\,m^{-2}}$ during the thaw season is primarily controlled by the date of its onset, i.e. date of snow melt-out, and secondarily by the weather throughout the thaw season. Second, convective heat transfer selectively enhances cooling over warming (thermal semi-conductor effect) as shown by time-varying effective thermal conductivity that increase from $1.2\,\mathrm{W\,m^{-1}\,K^{-1}}$ under strongly stable AL temperature gradients (weak warming) to episodically over $10\,\mathrm{W\,m^{-1}\,K^{-1}}$ under unstable AL temperature gradients (strong cooling). The snow cover controls whether at all and which type of buoyancy-driven cooling convection takes place: First, *Rayleigh ventilation* typically occurs in autumn when the atmosphere cools faster than the AL and air density instabilities induce convective overturning. It is the most efficient cooling process with episodically large, but short-lived upward fluxes up to $\sim 20\,\mathrm{W\,m^{-2}}$ at snow-free or snow-poor conditions. Second, beneath a semi-closed snow cover perforated by snow funnels, radiatively cooled air infiltrates into the AL. *Cold-air infiltration*/drainage leads to moderate, but persistent fluxes of $\sim 2-5\,\mathrm{W\,m^{-2}}$ that result in strong convective winter cooling in snow-poor winters. This cooling process is not diagnosed by Rayleigh numbers as the cold, dense air pools near the AL base, but should not be overlooked in future heat transfer modelling. Third, no convection occurred beneath a closed/insulating snow cover, small heat fluxes (within $2\,\mathrm{W\,m^{-2}}$) prevent a strong winter cooling.

Our field-based heat flux measurement and estimates of effective thermal conductivity $k_\mathrm{eff}$ are valuable for thermal numerical modelling. A thaw-season $k_\mathrm{eff}^0 = 1.2\,\mathrm{W\,m^{-1}\,K^{-1}}$ under stagnant (no convection) conditions indicates that radiative heat exchange is an important heat transfer process in coarse blocky material. This finding, which agrees with geotechnical laboratory experiments with crushed-rock beds, has often been overlooked in the geomorphological literature, although it tends to counteract undercooling. In the strongly ventilated near-surface AL, atmospheric wind and penetrating warm air tends to enhance mechanical turbulence and increase $k_\mathrm{eff}$ (*wind-forced convection*), leading to a thaw-season averaged $\bar{k}_\mathrm{eff} = 3\,\mathrm{W\,m^{-1}\,K^{-1}}$. A Stefan parametrisation with this field-measured $k_\mathrm{eff}$ successfully simulated the seasonal ground ice melt as measured in a nearby rock glacier furrow. Our measurement experience could guide future quantitative research and our derived values could

calibrate or validate numerical modelling studies like Renette et al. (2023) or Zegers et al. (2024). Mountain permafrost is entering uncharted territory where empirical relations based on past experience might no longer apply. Our study is a step towards process-based numerical modelling of coarse blocky landforms needed to anticipate their response to climate change.

*Data availability.*   The PERMOS data can be obtained from the PERMOS network (http://www.permos.ch), and the PERMA-XT measurement data from https://www.permos.ch//doi/permos-spec-2023-1 (doi:10.13093/permos-spec-2023-01).

## Appendix A: Electrical heat transfer analogy and local thermal (non-)equilibrium (LTE/LTNE)

A question that arose (Fig. 9) is why the $Q_{\mathrm{CGR3}}^{rad}$ heat flux is correlated with daily-average AL *air* temperature gradients, even though the thermal radiation is emitted by the rock surfaces, not by the air, and why the correlation deteriorates when taking hourly or 10-minute data (Fig. A1a), or when taking near-surface HFP/2 data (Fig. A1b). After all, even moist air is virtually transparent to thermal radiation at length scales encountered in the AL pore space. The question is of practical relevance because AL temperatures are more conveniently measured in the pore space rather than inside the blocks. This perhaps puzzling observation can be explained by a thermal resistance circuit (an electrical analogue to the heat transfer) and local thermal (non-)equilibrium (LTE/LTNE) which is important to understand the measurements in porous media consisting of constituents with diverging thermal inertia, here rock particles and air.

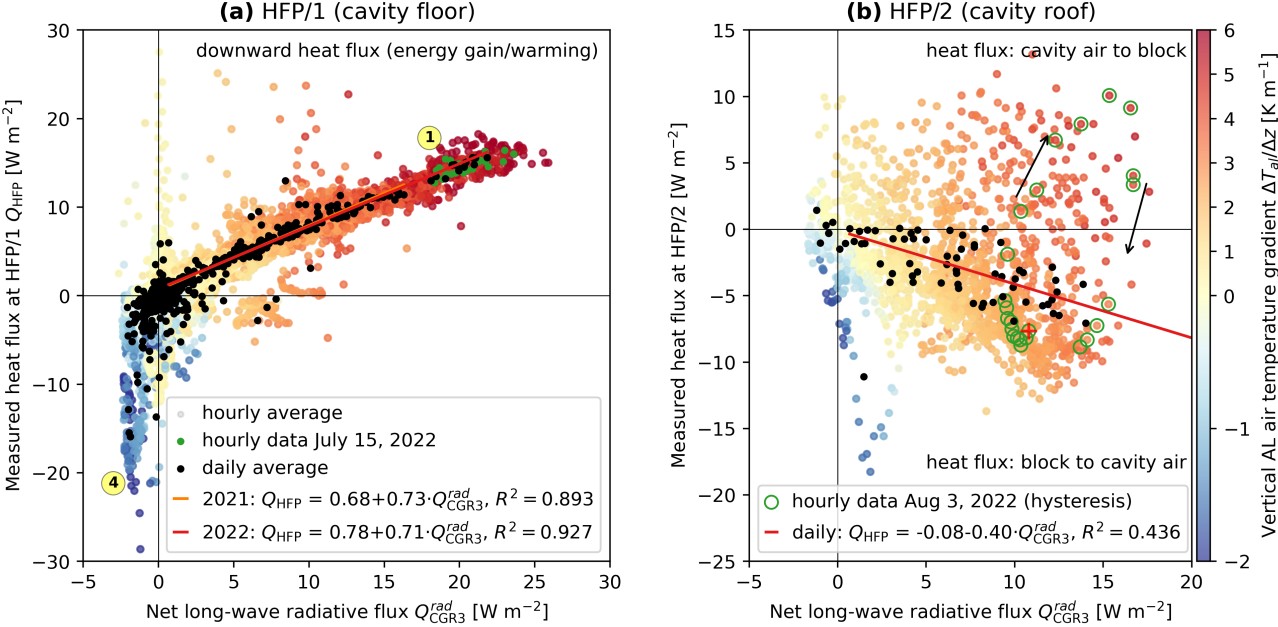

**Figure A1.** The heat flux plate measurements of HFP/1 ($Q_{\mathrm{HFP}}$) and HFP/2 (not discussed in main text) are correlated with the thermal net radiation $Q_{\mathrm{CGR3}}^{rad}$, but only for stable temperature gradients $\nabla_z T_{al}$ ($Q_{\mathrm{HFP}} \propto Q_{\mathrm{CGR3}}^{rad}$). Diurnal loops are stronger in the **(b)** HFP/2 data closer to the surface (at $-1.1$ m) because transient effects of the daily solar cycle are more intense than at **(a)** HFP/1 depth (at $-2.0$ m). At HFP/2, hourly heat fluxes shift direction, with heat moving upwards from the cavity into the block during warm afternoons in the thaw season, opposite of the downward daily-average heat fluxes (black dots). Sign convention: positive means into the block (downward for HFP/1, upwards for HFP/2; Fig. 3). Note the different y-axis: HFP/2 measurement available only after Jul 26, 2022.

In the unfrozen coarse blocky AL, the total heat transfer is composed of two "chains", air convection (turbulent fluxes $Q_h + Q_{le}$) in the pore space *parallel* to the heat transfer by the blocky matrix (Fig. A2). Heat transfer in the blocky matrix is composed of the heat conduction within the blocks $Q_c$ and radiative heat transfer $Q_r$ between blocks across the air-filled pore

space (air is transparent to thermal radiation on the pore length scale). Since the blocks barely touch (clast-supported/open-framework), particle-to-particle conduction is negligible and thermal radiation is the only heat transfer that links the blocks. Conduction and radiation operate *in series* (implying that $Q_r = Q_c$). Over timescales where a local thermal equilibrium (LTE) between air and blocks is reached ($\sim 1\,\mathrm{day}$) and in periods where convection does not dominate the heat transfer, locally uniform temperatures in the blocks and the air can be assumed. The different phases are no longer distinguished and the entire coarse blocky AL is treated as an effective medium having a single temperature profile represented by the below-ground air temperature $T_{al}$ that is more conveniently measured in the pore space (TK1) than inside the blocks (TK6). The overall heat transfer is treated as diffusive, which is true for conduction, applicable for thermal radiation in porous media (Fillion et al., 2011), but questionable for convection, and described by the effective thermal conductivity $k_{\mathrm{eff}}$ that lumps together all three conductive, radiative, and convective processes in both "chains". Hence, such a $k_{\mathrm{eff}}$ is only meaningful at LTE timescales. As indicated by the relation between $Q_{\mathrm{HFP}} \approx Q_G$, $Q_{\mathrm{CGR3}} \approx Q_r$, and $\nabla_z T_{al}$ valid for daily average values (minimum LTE timescale), the total heat flux $Q_G$ during most of the thaw season is represented by radiation $Q_r$ and the measured $Q_{\mathrm{CGR3}}$, $Q_r \approx Q_G$. At sub-daily timescales or during strong convection events, LTE is no longer a valid assumption. This is shown by diverging AL air (TK1) and rock (TK6) temperatures and the different hourly pattern of the $Q_{\mathrm{HFP}}$ and $Q_{\mathrm{CGR3}}$ measurements ($Q_{\mathrm{HFP}} \napprox Q_{\mathrm{CGR3}}$; Appendix Sect. B). Due to the thermal inertia of the blocks, rock temperatures and radiative heat transfer $Q_{\mathrm{CGR3}}$ lag behind AL air temperature and heat flux $Q_{\mathrm{HFP}}$. The total heat transfer is then adquately described by the local thermal non-equilibrium (LTNE) approach with phase-specific energy equations that account for the air–rock interface heat transfer (Marchenko, 2001; Zegers et al., 2024). The closer we look at the measurements gathered in the pore space of the AL, the more convective LTNE processes appear in the data.

## Appendix B: Sub-daily measurements reveal wind-forced convection

In the large and highly permeable instrumented main cavity, wind-forced convection transfers some heat to large AL depths $\sim 2\,\mathrm{m}$ even under stable air stratification and increases the heat transfer rate compared to radiation–conduction alone (Sect. 6.3.1). Sub-daily data show the processes (Fig. B1) and show the link between above- and below-ground conditions: Driven by the anabatic atmospheric wind (a thermal upslope wind that develops in the wind-sheltered cirque), AL airflow speeds are highest in the afternoon (Fig. B1b), precisely when the near-surface AL is most strongly heated and temperature gradient are largest (Fig. B1a). The (comparatively) strong afternoon winds counteract the stabilising positive temperature gradients. Warm air masses penetrate the permeable coarse blocky AL (shown by the afternoon HFP/2 measurements that indicate a heat flux *upwards into the block*, Fig. A1b). Forced convection transfers the heat downwards in the late morning–afternoon *parallel* (electrical analogue in Appendix Sect. A) to the radiative–conductive "background flux" (as shown by the TK1 and HFP/1 $Q_{\mathrm{HFP}}$ data; Fig. B1a, d), to which AL rock temperatures and the AL thermal net radiation $Q_{\mathrm{CGR3}}^{rad}$ respond to with some time lag (TK6/2 and $Q_{\mathrm{CGR3}}^{rad}$ peak in the evening; Fig. B1a, d). This pattern of atmospheric wind speed, AL airflow speed, and AL air temperature gradients that co-vary in phase is in turn an effect of the low-albedo debris surface (micro-topography) in the sheltered cirque (macro-topography) that gives rise to insolation-driven diurnal cycles. Such daily oscillations of the AL air

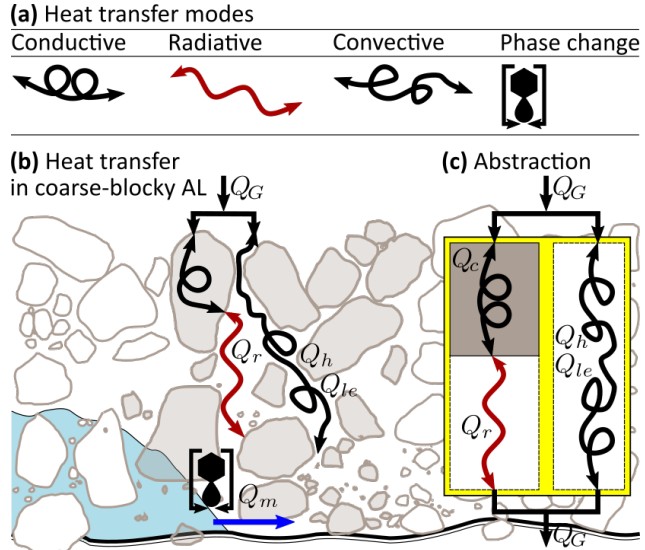

**Figure A2.** Schematic heat transfer in a dry unfrozen coarse blocky AL conceptualized as a resistance circuit. **(a)** Heat transfer modes (Sect. 4.1). **(b, c)** The total heat transfer $Q_G$ arises from convection $Q_h + Q_{le}$ (in the pore space) in parallel with radiation–conduction, $Q_G = Q_h + Q_{le} + Q_r$. Radiation–conduction is radiation $Q_r$ in pore space and conduction $Q_c$ in blocks in series. Figure inspired by Schneider (2014).

and rock temperatures without time lag down to $2.9$ m that indicate non-conductive heat transfer were also observed by Herz (2006) in the *Ritigraben* block slope.

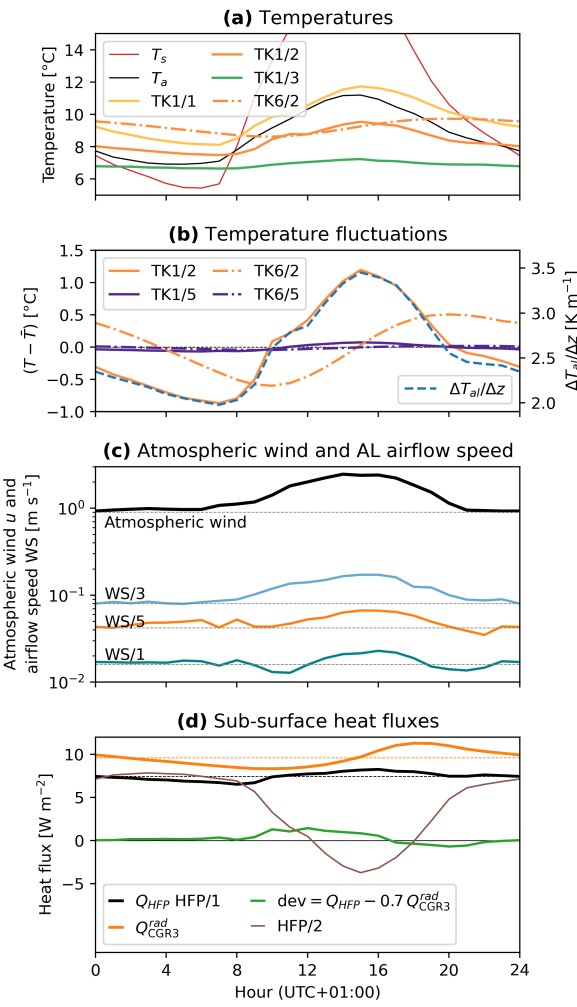

**Figure B1.** Evidence for wind-forced convection from sub-daily data: August 2022 hourly averages of **(a)** temperatures ($T_s$, $T_a$, AL air TK1, AL blocks TK6), **(b)** temperature fluctuations $T' := T - \bar{T}$ (24-h running mean subtracted) and gradient $\Delta T_{al}/\Delta z$, **(c)** AL airflow and wind speeds, and **(d)** measured AL heat fluxes. **(a, b)** AL air temperatures (TK1, —) and **(c)** AL airflow speeds down to 2.9 m (WS/1) show a daily course without time lag, only attenuated in amplitude. Rock temperatures (TK6, ⋯) lag behind AL air temperatures. **(d)** $Q_{\mathrm{HFP}}$ HFP/1 is in phase with airflow speed and AL air temperature gradient, whereas $Q_{\mathrm{CGR3}}^{rad}$ is in phase with the lagging rock temperatures TK6/2.

## Appendix C: Simple relations for modellers

We found simple relations between the below-ground radiative heat transfer and ice melt rates during the thaw season and the (remotely measureable) ground surface temperature. The numbers are site-specific, slightly differ between the thaw seasons 2021 and 2022, and are far from exact. The relations should be used with caution and are certainly not valid on timescales shorter than a few days. We nonetheless report them here because they suffice for rough order-of-magnitude estimates and potentially lead to simple tools useful for remote sensing and modelling applications.

First, the radiative–conductive downwards heat transfer can be related to the surface temperature $T_s$ on snow-free ground. Daily average in-cavity thermal net radiation $Q_{CGR3}^{rad}$ is correlated with the 2-m air $T_a$ ($R^2 = 0.738$ and $0.614$ for 2021 and 2022, respectively; plot not shown) and the radiometric ground surface temperature $T_s$ (derived from the PERMOS outgoing long-wave radiation, Amschwand et al. (2024a)) as long as $T_s$ is above the freezing point (Fig. C1). The correlation slightly improves for $T_s$ of the *previous day* rather than $T_s$ of the same day (2022 $R^2$ increases from $0.723$ to $0.786$). Like $\bar{k}_{eff}^{rad}$, also the $Q_{CGR3}^{rad}$–$T_s$ relation differs for the two thaw seasons 2021 and 2022, possibly due to the differing impact of convection that affects $T_s$ and $\nabla_z T_{al}$ (note that the $Q_{HFP}$–$Q_{CGR3}^{rad}$ relation is identical for both thaw seasons; Fig. A1a). Below 0°C and beneath snow-covered ground, the $Q_{CGR3}^{rad}$–$T_s$ relation breaks down and radiative fluxes remain small, within $\pm 2\,\mathrm{W\,m^{-2}}$, with $Q_{CGR3}^{rad}$ magnitude and direction that is independent of the outside air or surface temperatures.

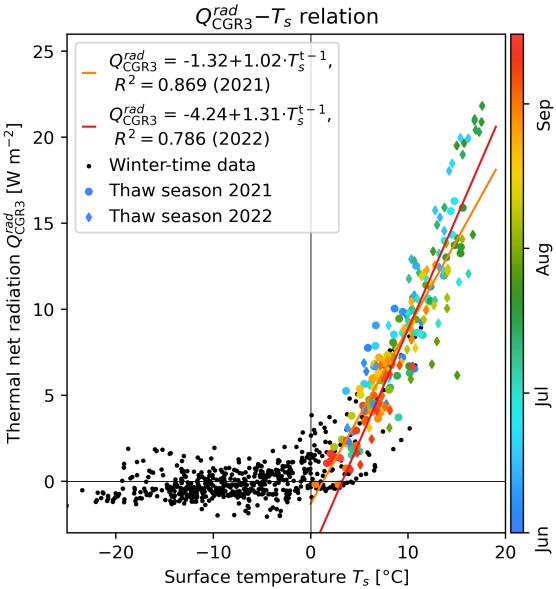

**Figure C1.** Thermal net radiation and ground surface temperature of the previous day are correlated beneath snow-free ground.

Second, a parameter used for modelling sub-debris melt rates on debris-covered glaciers is the thermal resistance. Assuming steady state conditions, an effective thermal resistance $R_{eff}$ [K m$^2$ W$^{-1}$] of the AL can be derived from the observed linear

temperature profile (Fig. 9) and the linear $Q_{\mathrm{CGR3}}^{rad}$–$T_s$ relation (Fig. C1; e.g., Nakawo and Young (1981, 1982); Kayastha et al.
(2000); Mihalcea et al. (2006); Fujita and Sakai (2014); Rounce and McKinney (2014)),

$$R_{\mathrm{eff}} := \frac{h_{al}}{\bar{k}_{\mathrm{eff}}^{rad}} = \frac{T_s - 0°\mathrm{C}}{Q_{\mathrm{CGR3}}^{rad}}, \tag{C1}$$

where $h_{al}$ is the AL thickness ($\sim 4$ m, extrapolated from the linear temperature profiles). The inverse thermal resistance corresponds to thermal conductivity normalized by AL thickness. Both formulations yield similar values of $R_{\mathrm{eff}} \approx 1.0 \pm 0.2$ K m$^2$ W$^{-1}$.

Third, also the linear regression of the stake measurements (converted to melt heat flux $Q_m$ using Eq. 7) with the ground surface temperature $T_s$ yields $Q_m = -0.1 + (0.7 \pm 0.2)T_s$ [W m$^{-2}$] (Fig. 12d), which is consistent with above $R_{\mathrm{eff}}$ derived from the radiation measurements (Eq. C1, taking $Q_m = Q_r$).

## Appendix D: Notes on upscaling: Variability of intrinsic permeability $K$ and radiative thermal conductivity $k^r$

The contribution of non-conductive heat transfer by air convection and thermal radiation is conditioned by the intrinsic permeability $K$ that generally increases with block/pore size. Here, we give quantitative formulae how the two related key parameters, the intrinsic permeability $K$ and the radiative thermal conductivity $k^r$, increase with effective particle size. The strong sensitivity of $K$ and $k_{\mathrm{eff}}^0 := k^r$ to debris texture at landform scale needs be kept in mind when attempting to upscale from plot-scale measurements.

### D1  Intrinsic permeability $K_{KC}$

The intrinsic permeability $K$, an indication of the ability for fluids to pass through the porous medium, is commonly estimated via the the Kozeny–Carman relation (Wicky and Hauck, 2020)

$$K_{KC} = \frac{1}{4.25} \frac{\phi_{al}^3}{5(6/d_{10})^2 (1 - \phi_{al})^2}, \tag{D1}$$

where $\phi_{al} = 0.4$ is the porosity, $d_{10}$ a characteristic grain diameter such that $10\,\%$ of the particles are smaller than $d_{10}$, and $1/4.25$ is the empirical Côté et al. (2011) correction factor for coarse material. For a characteristic block diameter $d_{10} = 0.3$ m, Eq. D1 predicts $\sim 4.7 \times 10^{-6}$ m$^2$, reasonably agreeing with the estimated $3 \times 10^{-6}$ K$^2$ by Wicky and Hauck (2020) inferred from thermal numerical modelling. Although the Kozeny–Carman relation has not been rigorously tested for Murtèl-sized debris composed of non-spherical blocks, estimates from different studies are consistent (Herz, 2006; Wicky and Hauck, 2020; Côté et al., 2011) and the Kozeny–Carman relation has proven useful even in turbulent airflow regimes far from Darcian. The Kozeny–Carman relation implies that the permeability $K$ scales with $d_{10}^2$, suggesting lateral and vertical variability even on the same landform, as fine-material is typically more abundant near the AL base.

### D2  Radiative thermal conductivity $k^r$

Radiative thermal conductivity increases with block/pore size (actually: the effective length for radiation between particles) and mean temperature cubed, $k_{\mathrm{eff}}^0 = k^r \sim (d_{10}, \ \sigma \bar{T}^3)$. The larger the pores and the distance between particles, the larger the

surface temperature differences across the pore space and the radiative thermal conductivity $k^r$ since the resulting radiative flux

$$Q_r = E\sigma(T_2^4 - T_1^4) \tag{D2}$$

is independent of the inter-particle distance (Fillion et al., 2011; Lebeau and Konrad, 2016). Radiative heat transfer bridges the pore space by bypassing the high conductive contact resistance between the blocks, whereas conduction transfers the heat within the blocks (Vortmeyer, 1979). Radiative heat transfer in porous media with opaque particles is effectively diffusive and along the temperature gradient, analogous to heat conduction. Hence, a radiative thermal conductivity $k^r$ analogous to a (conductive) thermal conductivity can be defined. The radiative conductivity is obtained from linearisation of Eq. D2 to recast it as a flux–gradient relation (diffusion equation) of the form $Q_r := k^r (\nabla_z T_{al})$ (using $(T_2^4 - T_1^4) \approx 4\bar{T}^3(T_2 - T_1)$, approximation valid for $(T_2 - T_1)/\bar{T} \ll 1$) (Kaviany, 1995; Lebeau and Konrad, 2016; Esence et al., 2017; Rieksts et al., 2019),

$$k^r = 4Ed_{10}\sigma\bar{T}^3. \tag{D3}$$

$E$ is a semi-empirical exchange factor (that absorbs the surface emissivity $\varepsilon$, the rock thermal conductivity $k_r$, and accounts for the particle arrangement), $d_{10}$ the effective particle diameter (10% of the whole material mass has particles smaller than $d_{10}$, Fillion et al. (2011)), $\sigma$ the Stefan-Boltzmann constant, and $\bar{T} := (T_1 + T_2)/2$ a characteristic mean temperature. Note that radiative heat transfer counteracts undercooling because $k^r$ increases with temperature, i.e., at higher temperatures (during the thaw season), more heat is transferred under the same absolute temperature gradient (Fillion et al., 2011). This "radiative asymmetry" is opposite to the convective thermal semi-conductor effect. Using the semi-empirical Eq. D3 (Eq. 11 in Fillion et al. (2011)) yields $1.0 \leq k_{eff}^0 \leq 2.6$ W m$^{-1}$ K$^{-1}$ for $0.3 \leq d_{10} \leq 0.8$ m ($E := \varepsilon/(2-\varepsilon)$, $\varepsilon = 0.8$, $\bar{T} = 5°$C, $\sigma = 5.67 \times 10^{-8}$ W m$^{-2}$ K$^{-4}$). As for $K_{KC}$, beware of extrapolation: Due to the increasing thermal resistance, $k^r$ no longer scales linearly for large blocks ('particle non-isothermality effect', Singh and Kaviany, 1994; Ryan et al., 2020).

## Appendix E:  Seasonal patterns and drivers of air circulation

The seasonal airflow speed pattern controlled by the snow cover is shown in Fig. E1. At large AL depth (deepest WS/1 at $-2.1$ m, Fig. E2a), airflow speed were highest at unstable air density stratification (Rayleigh ventilation) and isothermal cavity at high outside wind speeds (wind-forced convection). Atmospheric wind set the labilized air column down to the cavity base in motion. At stable AL air temperature gradients, airflow speed was overall low, but even then, airflow speeds tended to be higher under high atmospheric wind speed. The effect of wind-forced convection was weak, but detectable in the wide instrumented cavity down to 2 m depth. Note the striking similarity with Fig. A1a. Near the surface (WS/6 in Fig. E2b–c), airflow speed was overall higher under snow-free conditions, increased with atmospheric wind speed (wind-forced ventilation), and was insensitive to the (anyway mostly stable) vertical temperature gradient. The thicker the snow cover and the stronger the decoupling between AL and atmosphere (AL–atmosphere coupling in Amschwand et al. (2024a)), the more important density contrasts became to drive the air circulation (buoyancy-driven ventilation), however at overall lower airflow speeds (Fig. E2b).

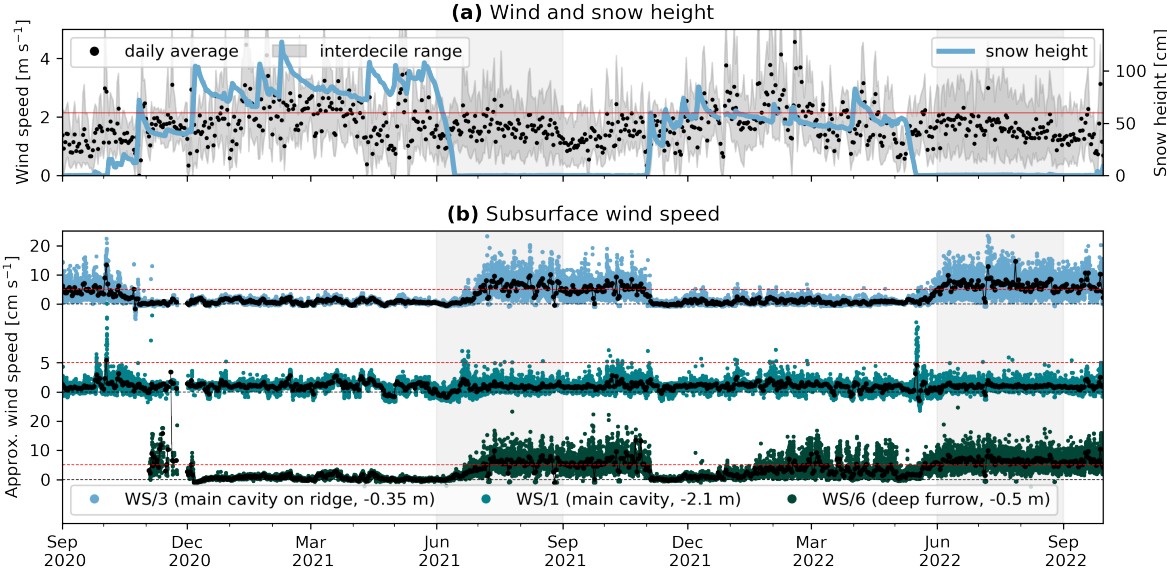

**Figure E1. (b)** Sub-surface airflow speed (WS) measurements with **(a)** wind speed and snow for context.

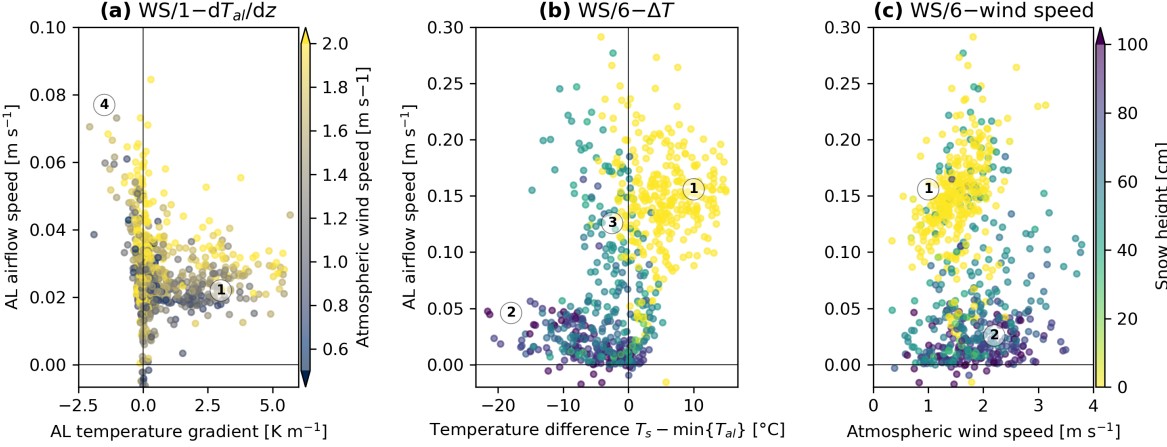

**Figure E2.** Drivers of ventilation. **(a)** Ventilation at depth (WS/1) is primarily buoyancy-driven (④). At stable stratification (positive temperature gradients), airflow speeds are low but enhanced by the atmospheric wind (①). **(b, c)** The near-surface ventilation (WS/6) transitions from mainly wind-driven (①) to buoyancy-driven circulation with increasing snow height (②, ③). The circled numbers ①–④ refer to Table 3.

## Appendix F: Literature values for apparent thermal diffusivity $\kappa_a$

Literature values for the apparent thermal diffusivity $\kappa_a$ in periglacial landforms, supraglacial debris, and cryic regosol are listed in Table F1.

**Table F1.** Literature values for the apparent thermal diffusivity $\kappa_a$.

| Value $\kappa_a$ [$m^2$ $s^{-1}$] | Landform/context | Reference |
|---|---|---|
| $\sim 2.3 \times 10^{-6}$ | *Murtèl* rock glacier (ventilated) | this study |
| $9.6 \times 10^{-7}$ | *Murtèl* rock glacier (stagnant) | this study |
| $\sim 10^{-2} - 10^{-5}$ | *Murtèl–Chastelets* periglacial area | Hanson and Hoelzle (2005) |
| $2.7 \times 10^{-6}$ | *Chastelets* AL | Schneider et al. (2012) |
| $(1.3 - 1.6) \times 10^{-6}$ | *Chastelets* PF | Schneider et al. (2012) |
| $1.6 \times 10^{-6}$ | *Murtèl* bedrock | Schneider et al. (2012) |
| $\sim 10^{-3} - 10^{-4}$ | *Ritigraben* block slope | Herz (2006) |
| $2 \times 10^{-7}$ | *Juvvasshøe* block field (AL) | Isaksen et al. (2003) |
| $(0.5 - 2.0) \times 10^{-5}$ | openwork block field (stagnant) | Juliussen and Humlum (2008) |
| $\sim 7 \times 10^{-5}$ | openwork block field (ventilated) | Juliussen and Humlum (2008) |
| $(6.0 - 9.0) \times 10^{-7}$ | *Khumbu* debris covered glacier | Conway and Rasmussen (2000) |
| $(9.50 \pm 0.09) \times 10^{-7}$ | *Ngozumpa* debris-covered glacier | Nicholson and Benn (2013) |
| $(6.41 \pm 2.21) \times 10^{-7}$ | *Lirung* debris-covered glacier | Steiner et al. (2021) |
| $(1.6 - 2.0) \times 10^{-7}$ | cryic regosol | Mendoza López et al. (2023) |

AL = active layer; PF = permafrost body. Cf. $k_{eff}$ for supraglacial debris tabulated in Rowan et al. (2021).

## Appendix G: Nomenclature

Variables, parameters and constants used in this study are tabulated in Table G1.

**Table G1.** Nomenclature: Measurement variables, parameters, dimensionless numbers, and constants.

| Symbol | Unit | Name | Symbol | Unit | Name |
|---|---|---|---|---|---|
| $C_p$ | J kg⁻¹ K⁻¹ | Isobaric specific heat capacity of moist air | $T_s$ | K, °C | Surface temperature |
| $C_v$ | J m⁻³ K⁻¹ | AL volumetric heat capacity | | | (coarse-blocky AL, snow surface) |
| $c_w, c_r$ | J kg⁻¹ K⁻¹ | Specific heat capacity of water, rock | $T_a, T_{wb}$ | K, °C | Air temperature (dry-bulb, wet-bulb) |
| $d_{10}$ | m | Effective particle diameter | $T_{al}$ | K, °C | AL temperature |
| $E$ | 1 | Semi-empirical radiation exchange factor | $T_{Pr}$ | K, °C | Rainwater temperature |
| $f_i\ (f_1, f_2)$ | m³ m⁻³ | (Layer-wise) volumetric ice content | $T_r$ | K, °C | Temperature in blocks |
| $H_{al}^{\theta}, \Delta H_{al}^{\theta}$ | J m⁻², W m⁻² | Sensible heat storage (change) | $T_{CGR3}$ | °C | Pyrgeometer housing temperature |
| $h_{al}, h_S$ | m | Thickness of coarse-blocky AL, snow cover | $\frac{\mathrm{d}T_{al}}{\mathrm{d}z} := \nabla_z T_{al}$ | K m⁻¹ | Vertical AL temperature gradient |
| $h_{\mathrm{WS}}$ | W m⁻² K⁻¹ | WS01 heat transfer coefficient | $t$ | s, h, d | Time |
| $I_t$ | °C × d | Surface thaw index | $u$ | m s⁻¹ | Wind or airflow speed |
| $K, K_{KC}$ | m² | Intrinsic AL permeability (estimated | $u_{\mathrm{WS}}$ | m s⁻¹ | WS01 airflow speed |
| | | with the Kozeny–Carman equation Eq. D1) | $z$ | m | Vertical coordinate |
| $k_{\mathrm{PF}}$ | W m⁻¹ K⁻¹ | Thermal conductivity of permafrost body | $\beta_a \approx 1/T_0$ | (273 K)⁻¹ | Air thermal expansion coefficient |
| $k_{\mathrm{eff}}, k_{\mathrm{eff}}^{0}$ | W m⁻¹ K⁻¹ | AL (stagnant) effective thermal conductivity | $\varepsilon$ | 1 | Surface emissivity (snow, blocky surface) |
| $\bar{k}_{\mathrm{eff}}^{rad}$ | W m⁻¹ K⁻¹ | Pyrgeometer-derived average $k_{\mathrm{eff}}$ | $\zeta$ | m | Depth of ground-ice table |
| $k^r$ | W m⁻¹ K⁻¹ | Radiative thermal conductivity | $\kappa_a, \bar{\kappa}_a$ | m² s⁻¹ | Apparent thermal diffusivity |
| $k_r$ | W m⁻¹ K⁻¹ | Rock thermal conductivity | | | (thaw season log-mean average) |
| $L_{al}^{\downarrow}, L_{al}^{\uparrow}$ | W m⁻² | Down-/upwards thermal radiation | $\mu_a$ | Pa s | Air dynamic viscosity |
| $Q_G$ | W m⁻² | Ground heat flux | $\rho_a, \rho_r, \rho_i$ | kg m⁻³ | Density of air, rock, ice |
| $Q_h, Q_{le}$ | W m⁻² | Sensible and latent turbulent flux within AL | $\phi_{al}$ | 1 | Porosity of coarse-blocky AL |
| $Q_m$ | W m⁻² | Ground-ice melt heat flux | *Dimensionless numbers* | | |
| $Q_r$ or $L$ | W m⁻² | Radiative heat flux (thermal radiation) | $Ra$ | 1 | Rayleigh number (Eq. 3) |
| $Q_{\mathrm{CGR3}}^{rad}$ | W m⁻² | Pyrgeometer net measurement | *Constants* (value) | | |
| $Q_{\mathrm{HFP}}$ | W m⁻² | Heat flux plate measurement | $g$ | m s⁻² | Gravitational acceleration (9.81) |
| $Q_{\mathrm{PF}}$ | W m⁻² | Permafrost heat flux | $L_m$ | J kg⁻¹ | Latent heat of melting ($3.34 \times 10^5$) |
| $q, q^*$ | g g⁻¹ | Specific humidity (at saturation) | $\sigma$ | W m⁻² K⁻⁴ | Stefan–Boltzmann constant |
| $q_a$ | g g⁻¹ | Air specific humidity (2 m above ground) | | | ($5.670 \times 10^{-8}$) |
| $q_{al}$ | g g⁻¹ | AL specific humidity | | | |
| $R_{\mathrm{eff}}$ | K m² W⁻¹ | Effective thermal resistance (Eq. C1) | | | |
| $r$ | m³ m⁻² s⁻¹ | Rainfall rate | | | |
| rH | % | Relative humidity | | | |
| SWE | kg m⁻² | Snow water equivalent | | | |

*Author contributions.* DA performed the fieldwork, model development and analyses for the study and wrote the manuscript. MS, MH and BK supervised the study, provided financial and field support and contributed to the manuscript preparation. AH and CK provided logistical support and editorial suggestions on the manuscript. HG designed the novel sensor array, regularly checked data quality, contributed to the analyses and provided editorial suggestions on the manuscript.

*Competing interests.* The authors declare that they have no conflict of interest.

*Acknowledgements.* This work is a collaboration between the University of Fribourg and GEOTEST and was funded by the Swiss Innovation Agency Innosuisse (project 36242.1 IP-EE 'Permafrost Meltwater Assessment eXpert Tool PERMA-XT'). The authors wish to thank Walter Jäger (Waljag GmbH, Malans) and Thomas Sarbach (Sarbach Mechanik, St. Niklaus) for the technical support, the Corvatsch cable car company for logistical support, and Marc Lütscher (SISKA, La Chaux-de-Fonds) for the discussions.

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
