# Peer review of "Sub-surface processes and heat fluxes at coarse blocky Murtèl rock glacier (Engadine, eastern Swiss Alps): Seasonal ice and convective cooling render rock glaciers climate-robust"

_EGUsphere, 2024_

## Referee Comment (RC2)

Review of:

**Sub-surface processes and heat fluxes at coarse-blocky Murtèl rock glacier (Engadine, eastern Swiss Alps)**

Submitted to: *Earth Surface Dynamics*
Reviewer: Benjamin Hills

**Summary.**

Amschwand et al. present a new and unique dataset which constrains the thermodynamics of a rock glacier in the Swiss Alps. To interpret their data, they also design thermodynamic models for heat fluxes in and out of the rock glacier system. I believe this dataset is of great interest to the readership of *Earth Surface Dynamics*, and after revisions I believe the article will fit well in the journal. As written, I worry that the importance of this work could be lost in the complexity of its presentation. Most of my requests below aim to help refine the narrative of the article and to elevate its impact.

**General Comments.**

I believe you are underselling your work by focusing on details and skipping over some of the high-level impact/importance of your work on rock glaciers. For instance, there is no introductory statement (not in the abstract or introduction section) to explain *why* the reader should care about rock glaciers to begin with. The final statement in the conclusion effectively states that "more work needs to be done" which could leave a reader confused on what they are meant to take away from your article. Even the title could be changed to increase the interest in the article, bringing in something about the climate resiliency or the 'semiconductor' effect that you mention (see my comment on that below). I do realize you want to keep consistency with the title in your previous article (Amschwand et al., 2024), so maybe it is too late for that change.

Some of the analysis is thorough to the point of redundancy. I do appreciate the full exploration of every aspect of the data, but I feel that much of it could be moved to a supplementary text for the sake of preserving the clearest narrative possible in the main article. Some examples:
- Thermal conductivity, thermal diffusivity, and thermal resistance are all presented as material properties of interest. I understand that each is slightly different, but they are all related and trying to get at the same thing. That is, "how long does it take to heat up or cool down the active layer". I would focus on one material property within the main text (probably the thermal diffusivity since it contains the conductivity and since I believe Figure 10 is one of the more important takeaways). Then, move any other discussion to the supplement.
- There are *many* variables, and many different fluxes to keep track of. I suggest a table to list all the notation.
- The two temperature models, degree-day model and Stefan model, give effectively the same result (e.g., in figure 13 a and b). I argue it is not worth the confusion it adds to

explain each of them separately. Instead, just use the model which is best suited to your narrative (probably the Stefan model in my opinion).
- The data presented in the appendix is not critical to the main article and it is not presented as a significant component of the study. I believe these figures could be moved to a supplement which would reduce the length of the article with no drawback to the narrative.
- How important are the "short-lived events" to the overall energy balance of the rock glacier? My sense is that this is additional information which could be kept to the supplement (e.g. move figures 15 and 16 there) to leave the reader with the most important takeaways (e.g. I think that Figure 14 is quite important).

Most of the measurements critical to the interpretation presented in the article are from a single subsurface cavity. It is not clear that this single cavity is representative of the AL over the entire area or especially to other rock glaciers. Some explanation within the Discussion section (even if speculative) could help the reader to understand whether the measurements made there can be extended away from the one cavity.

**Specific Comments.**

Abstract – I believe the second thaw-season mechanism is the most new and interesting result in the entire study. I would like to see it elevated and described more clearly in the abstract. As it is written, it is not clear to the reader that the 1.2 W/m/K vs. the 10 W/m/K are in opposite directions (i.e., that the increased conductivity actually acts to cool down the AL which is perhaps counterintuitive and a strong result). You mention the "thermal semiconductor" here in regard to this as well which I do not believe is explained in the text. Perhaps this is a known phenomena, but I had never heard of it and I think it is a useful analogy to talk about your system. Please discuss this semiconductor analogy more clearly in the text if you are going to use that term in the abstract.

L16 - Intro gets into detail very quickly. Maybe one more sentence to describe what the coarse-blocky active layer is in plain language?

L21 – I believe a comma after "Snow" would help readability.

L 31 – Not sufficiently clear yet what the term "undercooling" is referring to.

L70-82 – This paragraph feels more like methods to me.

L99 – Missing unit on the "2"

L163 and 180 – I think either make these subsections or just make it clear in the paragraph what you are talking about.

L170 – I am confused about this Nu parameterization, is it used later? If not, I think it only adds confusion to define it here.

L174 – Have you defined the Richardson number?

L197 - I would move all of section 4.1 into the Flux section 4.3. What is currently section 4.2 can be the explanation in words and then what is currently 4.3 serves to explain all the variables in a mathematical context.

L286 – seems to be a missing word? "on melting ground ice the AL"

L289 – citation for the Stefan equation. Also, could be worth stating in words that this is sensible/latent heat comparison.

L309 – make it clearer that this is an update to eq 11 and drop the parenthetical unless it is seen as necessary.

L341 – is this the first time Tal has been defined? It was used above.

L343 – Do you mean Tal<0.5C? I don't see how temperatures close to the melting point would minimize latent heat exchange.

L354 – need a citation on the "rarely been occurring in the last ~15 years"

L402 – not sure what the exclamation mark is for.

L415 – I believe you mean Figure 8b.

L431 – I believe there is a typo with the log-mean variable "alpha_alpha", shouldn't that bee kappa?

**Figures**

Overall, the figures are beautiful, very nice aesthetic.

Figure 1 – Beautiful map. Can you give a projection for the Northing/Easting that you show on the x/y axes, I assume it is UTM? Please describe the red lines in the caption, Does the latitude line correspond to both the inset and the main map? Unclear.

Figure 3 – I assume that the horizontal scale is equal to the vertical in the illustration. Is the same scale used in the "schematic horizontal section"? If not, a scalebar there would help.

Figure 4 – Shouldn't the t0 schematic go on the left with (a)?

Figure 5 – Indicate what the red and black contours are in the caption.

Figure 6 – Having the legends for b-d spread out through all the panels is confusing. I would make it one legend to the right of all the panels.

Figure 8 – Is the data on the x-axis the same for both (a) and (b)? If so, I am confused why the extents for the axes are different and why the net longwave extends into the range of 20-25 W/m2 for (a) but not for (b).

Figure 14 – I would remind the reader which of these fluxes are meant to sum together. That is, from equation (3) the purple, cyan, and black lines sum to equal green, but don't assume the reader will be able to remember eq (3). You do a good job of explaining this in the text 6.1.1, add some indicator either to the plot or the caption on how a reader should interpret this figure (perhaps redefining the flux terms in words instead of only the symbols, its hard to keep track of all of them).

Table 3 – This should be presented closer to where it is first introduced.

**References**

Amschwand, D., Scherler, M., Hoelzle, M., Krummenacher, B., Haberkorn, A., Kienholz, C., & Gubler, H. (2024). Surface heat fluxes at coarse blocky Murtèl rock glacier (Engadine, eastern Swiss Alps). *Cryosphere*, *18*(4), 2103–2139. https://doi.org/10.5194/tc-18-2103-2024

---

## Author Comment (AC1)

Dear editor Jens Turowski
Dear reviewers, Benjamin Hills and anonymous reviewer #1

We thank you for the constructive, careful and encouraging reviews. We appreciate your time and dedication to read through and comment on an admittedly long paper. We have thoroughly restructured the paper and updated figures along the lines suggested by the reviewers to emphasize topics of broad interest, while technical aspects have been moved to the Appendix. The major changes are:

- We have added discussion sections that are of general interest and have moved technical details to the Appendix. Added sections include:
    - We present a first quantitative, field-data based estimate of the climate sensitivity of rock glaciers, a debated topic that is poorly supported by quantitative data.
    - We better discuss the transferability of the Murtèl findings to other sites.
- We have streamlined the presentation of the data analysis by focusing on seasonal and daily time scales. Analyses at sub-daily (hourly) timescale are moved to the Appendix. The introduction has been rewritten to introduce why undercooling is relevant (the big picture), and the discussion has been strongly modified.
- We have updated the manuscript title to "Sub-surface processes and heat fluxes at coarse-blocky Murtèl rock glacier (Engadine, eastern Swiss Alps): Seasonal ice and convective cooling render rock glaciers climate-robust" to reflect the broader significance while keeping consistency with the previous publication.
- We have reduced the complexity of the topics discussed in the main text. Technical aspects like a thermodynamic heat transfer model, thermal non-equilibrium (LTE), the thermal resistance, and relations for the intrinsic permeability and radiative thermal conductivity are now in the Appendix. We do believe that while these aspects have distracted from the flow, they are based on a unique data set and worthwhile to be published.
- We have redrawn many figures as suggested by the reviewers. They are now better embedded in the text and better linked among themselves and to Table 3.

We hope that we address the reviewers' issues satisfactorily in the revised manuscript. In the following, we would like to reply to their comments and concerns in detail (reviewer comment in blue, >our response in black):

Reviewer #2: Benjamin Hills

Summary.

Amschwand et al. present a new and unique dataset which constrains the thermodynamics of a rock glacier in the Swiss Alps. To interpret their data, they also design thermodynamic models for heat fluxes in and out of the rock glacier system. I believe this dataset is of great interest to the readership of Earth Surface Dynamics, and after revisions I believe the article will fit well in the journal. As written, I worry that the importance of this work could be lost in the complexity of its presentation. Most of my requests below aim to help refine the narrative of the article and to elevate its impact.

General Comments.

I believe you are underselling your work by focusing on details and skipping over some of the high-level impact/importance of your work on rock glaciers. For instance, there is no introductory statement (not in the abstract or introduction section) to explain why the reader should care about rock glaciers to begin with. The final statement in the conclusion effectively states that "more work needs to be done" which could leave a reader confused on what they are meant to take away from your article. Even the title could be changed to increase the interest in the article, bringing in something about the climate resiliency or the 'semiconductor' effect that you mention (see my comment on that below). I do realize you want to keep consistency with the title in your previous article (Amschwand et al., 2024), so maybe it is too late for that change. Some of the analysis is thorough to the point of redundancy. I do appreciate the full exploration of every aspect of the data, but I feel that much of it could be moved to a supplementary text for the sake of preserving the clearest narrative possible in the main article. >We thank Benjamin Hills for these constructive comments and the examples suggested below, which we have included in the modified manuscript. Some examples:

- Thermal conductivity, thermal diffusivity, and thermal resistance are all presented as material properties of interest. I understand that each is slightly different, but they are all related and trying to get at the same thing. That is, "how long does it take to heat up or cool down the active layer". I would focus on one material property within the main text (probably the thermal diffusivity since it contains the conductivity and since I believe Figure 10 is one of the more important takeaways). Then, move any other discussion to the supplement. >We have retained two heat transfer parameters in the main text, the effective thermal conductivity and the diffusivity. Distinguishing these two parameters helps to avoid confusion, since they were derived independently from each other.
- There are many variables, and many different fluxes to keep track of. I suggest a table to list all the notation. >Yes, we have added such a list.
- The two temperature models, degree-day model and Stefan model, give effectively the same result (e.g., in figure 13 a and b). I argue it is not worth the confusion it adds to explain each of them separately. Instead, just use the model which is best suited to your narrative (probably the Stefan model in my opinion). >Yes, we have removed the degree-day model.
- The data presented in the appendix is not critical to the main article and it is not presented as a significant component of the study. I believe these figures could be moved to a supplement which would reduce the length of the article with no drawback to the narrative. >We would like to keep it in the Appendix for faster accessibility.
- How important are the "short-lived events" to the overall energy balance of the rock glacier? My sense is that this is additional information which could be kept to the supplement (e.g. move figures 15 and 16 there) to leave the reader with the most important takeaways (e.g. I think that Figure 14 is quite important). >The short-lived events, i.e. the Rayleigh convection in autumn, influence the energy budget of the entire winter. This is now better explained. Figures 15 and 16 are moved out of the main text.

Most of the measurements critical to the interpretation presented in the article are from a single subsurface cavity. It is not clear that this single cavity is representative of the AL over the entire area or especially to other rock glaciers. Some explanation within the Discussion section (even if speculative) could help the reader to understand whether the measurements made there can be extended away from the one cavity. >We agree and have added a section "Scope and transferability" in the Discussion section.

**Specific Comments.**

Abstract – I believe the second thaw-season mechanism is the most new and interesting result in the entire study. I would like to see it elevated and described more clearly in the abstract. As it is written, it is not clear to the reader that the 1.2 W/m/K vs. the 10 W/m/K are in opposite directions (i.e., that the increased conductivity actually acts to cool down the AL which is perhaps counterintuitive and a strong result). You mention the "thermal semiconductor" here in regard to this as well which I do not believe is explained in the text. Perhaps this is a known phenomena, but I had never heard of it and I think it is a useful analogy to talk about your system. Please discuss this semiconductor analogy more clearly in the text if you are going to use that term in the abstract. >We now state clearly that the thermal conductivity is sensitive to the direction of the thermal gradient and heat flow. Furthermore, we made the thermal semiconductor effect better visible in the discussion by mentioning it in the new section heading ("6.3.1, The thermal semiconductor effect: …").

L16 - Intro gets into detail very quickly. Maybe one more sentence to describe what the coarse-blocky active layer is in plain language? >We agree and have expanded the introduction to give the big picture first. Moreover, we have added: "clast-supported, coarse debris (dm-sized blocks, sparse fine material)".

L21 – I believe a comma after "Snow" would help readability. >The paragraph containing this sentence has been modified.

L 31 – Not sufficiently clear yet what the term "undercooling" is referring to. >We have rewritten the Introduction to provide the bigger picture what undercooling means.

L70-82 – This paragraph feels more like methods to me. >We have removed this paragraph.

L99 – Missing unit on the "2" >Thank you for spotting the type, unit has been added.

L163 and 180 – I think either make these subsections or just make it clear in the paragraph what you are talking about. >We have removed the paragraph headers.

L170 – I am confused about this Nu parameterization, is it used later? If not, I think it only adds confusion to define it here. >The Nusselt parameterization is only used in the context of the WS01 measurements.

L174 – Have you defined the Richardson number? >The (unneeded) Richardson number is no longer mentioned for clarity.

L197 - I would move all of section 4.1 into the Flux section 4.3. What is currently section 4.2 can be the explanation in words and then what is currently 4.3 serves to explain all the variables in a mathematical context. >Yes, we agree and have restructured these sections accordingly.

L286 – seems to be a missing word? "on melting ground ice the AL" >Yes, we have added "on".

L289 – citation for the Stefan equation. Also, could be worth stating in words that this is sensible/latent heat comparison. >We have added it.

L309 – make it clearer that this is an update to eq 11 and drop the parenthetical unless it is seen as necessary. >Yes, we have explained it. Parenthetical use is necessary because they represent the two layers.

L341 – is this the first time Tal has been defined? It was used above. >Yes, deleted here.

L343 – Do you mean Tal<0.5C? I don't see how temperatures close to the melting point would minimize latent heat exchange. >No, we use only values from the unfrozen AL Tal>0.5°C, the thaw season well after the zero curtain. We have made clearer by adding "latent effects of freezing/thawing".

L354 – need a citation on the "rarely been occurring in the last ~15 years" >These are observations made in our research group. We have been visiting the site multiple times per year in the framework of intense research and monitoring work in the past decades.

L402 – not sure what the exclamation mark is for. >Yes, deleted.

L415 – I believe you mean Figure 8b. >Yes, correct. Text modified accordingly.

L431 – I believe there is a typo with the log-mean variable "alpha_alpha", shouldn't that bee kappa? >Yes, correct. We have corrected the typo in the text and figures.

**Figures**

Overall, the figures are beautiful, very nice aesthetic. >Thank you.

Figure 1 – Beautiful map. Can you give a projection for the Northing/Easting that you show on the x/y axes, I assume it is UTM? Please describe the red lines in the caption, Does the latitude line correspond to both the inset and the main map? Unclear. >It is in the LV95 Swiss coordinate system (now mentioned in Fig. caption). Yes, red latitude line corresponds to both inset and main map (also mentioned now).

Figure 3 – I assume that the horizontal scale is equal to the vertical in the illustration. Is the same scale used in the "schematic horizontal section"? If not, a scalebar there would help. >Yes, the scale is the same. We've added a horizontal scale.

Figure 4 – Shouldn't the t0 schematic go on the left with (a)? >Yes, we have modified the drawing accordingly.

Figure 5 – Indicate what the red and black contours are in the caption. >Yes, contour lines are now mentioned in caption.

Figure 6 – Having the legends for b-d spread out through all the panels is confusing. I would make it one legend to the right of all the panels. >Yes, we agree. We have redrawn this figure as suggested by you and reviewer #1.

Figure 8 – Is the data on the x-axis the same for both (a) and (b)? If so, I am confused why the extents for the axes are different and why the net longwave extends into the range of 20-25 W/m2 for (a) but not for (b). >This figure is now in the Appendix, but not redrawn. The different axis extent is due to different data availability.

Figure 14 – I would remind the reader which of these fluxes are meant to sum together. That is, from equation (3) the purple, cyan, and black lines sum to equal green, but don't assume the reader will be able to remember eq (3). You do a good job of explaining this in the text 6.1.1, add some indicator either to the plot or the caption on how a reader should interpret this figure (perhaps redefining the flux terms in words instead of only the symbols, its hard to keep track of all of them). >Yes, we agree (also brought up by reviewer #1). We have completely redrawn this figure to make it better understandable.

Table 3 – This should be presented closer to where it is first introduced. >Yes, the table is now at the beginning of the Results section.

**References**

Amschwand, D., Scherler, M., Hoelzle, M., Krummenacher, B., Haberkorn, A., Kienholz, C., & Gubler, H. (2024). Surface heat fluxes at coarse blocky Murtèl rock glacier (Engadine, eastern Swiss Alps). *Cryosphere*, *18*(4), 2103–2139. https://doi.org/10.5194/tc-18-2103-2024

Reviewer #1

The manuscript presents the unique data set collected in a rock glacier using innovative approaches. I am not aware of similar data set collected anywhere in the world, and I believe that the study has a tremendous potential to make a highly unique and significant contribution to the scientific understanding of rock glaciers. The authors are commended for collecting this highly valuable data in a challenging environment. Unfortunately, however, the manuscript is poorly developed and hard to comprehend. The English language is grammatically correct, but it is impenetrable in many places due to the overloading of information. The manuscript seems to have more than 14,000 words in the main part, excluding the reference list and appendix. This is excessively long compared to a standard length of contemporary research articles (e.g., 8000-8500 words). I feel that the manuscript was submitted prematurely and could have been made much more interesting and useful. Instead of bombarding the reader with all the data the authors have observed, they could select those data that are essential and most interesting, and develop a coherent story around those data. For example, the authors could use their own abstract as a guidance in the process of selecting the data and writing a concise and well-developed manuscript. Personally, I find the data presented in Figures 7, 10, 13, and 14 most interesting. In the following, I will make specific comments and suggestions on the contents of the current manuscript, but the authors can ignore my suggestion if a particular content is dropped in a new manuscript. >We thank the anonymous referee for these constructive comments and suggestions. We agree that Figs. 7,10,13 and 14 are the most important ones, and have restructured the manuscript along these lines. We hope that with the major changes (outlined above), the revised manuscript is easier to follow.

SPECIFIC COMMENTS

Line 75. This statement is not consistent with commonly accepted definition of REV, which should have a statistically meaningful number of pores to be able to define REV-scale variables such as porosity and mean pore size. Please redefine the REV for this study. >We have deleted this (very technical) part.

Line 350. It will be useful to report the maximum (i.e. peak) snowcover thickness and date here, so the reader can get a sense of the winter condition.

Line 355. The surface meteorological conditions are described in another paper, but they need to be briefly presented in this paper for the reader, who may not have time to read the second paper. For example, I suggest that the authors add another panel showing daily mean temperature and weekly (or monthly) precipitation. >We have expanded the paragraph on meteorological conditions and included an additional figure showing the air temperature, precipitation, and snow height.

Line 357. Please add '(Eq. 4)' after the Rayleigh-Darcy numbers. >Yes, added.

Figurer 6a. Please include the legends for Ra = 27 and 40 in the graph instead of explaining them in the figure caption. The former is much easier for the reader than the latter. >Yes, done.

Line 359. Please explain the critical Ra in the main texts, not in the figure caption. >Yes, done.

Line 361. To help the reader understand this sentence more easily, it will be useful to present Figures 5 and 6 together, along with another panel showing air temperature and precipitation,

and possibly snow depth. >Yes, we have now put together Figs. 5 (ground temperature) and 6a (Rayleigh). Air temperature and precipitation are shown in (the new) Fig. 4.

Figure 6b-6d. In this graphs, the highest point is air temperature. It is not appropriate to connect the air temperature point with the first subsurface temperature point by a straight line, because the temperature profile in the boundary layer is not linear. Please come up with another way of demonstrating the data. >Yes, this is true. We have removed the lines connecting the surface with the 2-m air temperature. Correspondence is shown by the colors.

Figure 6b. These graphs are difficult to comprehend. For example, there are two red lines for 2022 but no explanation is given. Also, no explanation is given in the caption for Ts. Please improve the presentation of these graphs. >We have redrawn these plots with one common legend for all panels (also suggested by reviewer #2).

Line 368. Table 3 needs to be shown together with Figures 6b-6d so the reader can easily understand the setting of each line. I suggest that Figure 6a be presented as a panel in Figure 5, and Figures 6b-6d be presented together with Table 3. >Yes, we agree, and have redrawn these figures accordingly. Table 3 comes now earlier, along with Fig. 7. Thank you for this suggestion.

Line 372. Dec-Mar. This is shown as Dec-Apr in the figure legend. Please be consistent. Each occurrence of inconsistent information detracts the reader's thought process and makes the texts more difficult to follow. >Yes, it is in fact the Dec-Apr average. We have changed the text accordingly.

Line 375-376. I do not see 'striking asymmetry' in Figure 6b. The data indicated by circle-2 seems to show the same deviation for both high and low sides. Please clarify. >This is meant on seasonal scale: Winter profiles are near isothermal, while summer profiles show steep temperature gradients. We have changed the sentence to: "Note the striking asymmetry between near-isothermal winter and steep (2.0-2.8 K m$^{-1}$) thaw-season temperature profiles (asymmetric envelopes)".

Line 380. To demonstrate 'close to saturation', it is much better to show relative humidity. Unless the authors intend to use specific humidity to estimate vapour flux, please use relative humidity in this graph. >Yes, this is a good suggestion. We have added a panel showing relative humidity.

Line 399. This is only for the spring of 2021. Please clarify that. >No, it occurred in both springs, 2021 and 2022. We have now marked it better in Fig. 7.

Line 400. 'Inferred from'. This is not clear. Please explain it. >We have deleted this (technical) paragraph.

Line 440. I find this section interesting. Please consider focusing on this section and deleting or condensing other less interesting sections. >Yes, we agree (see major changes).

Line 470. This section is also interesting. >Yes, we agree (see major changes).

Line 480. The sub-section on temperature index model seems less relevant and interesting than the previous sections. >Yes, we have deleted the temperature index model.

Line 475 and 483. Please use a consistent symbol for the derivative of zeta. If the dot notation is used, please define the symbol at its first occurrence. >We ensured consistent notation.

Line 506. This sub-section is interesting as well. >We agree and have expanded this section.

Line 507. 'denoted by the summation symbol'. I do not see the summation symbol in Figure 14. Please use a consistent notation. >We have deleted the summation symbol.

Line 508. The AL exits the zero-curtain phase. Where in Figure 14 is this indicated? Please explain. >We agree that this was not shown. We now have added the zero curtains in the temperature and humidity profiles (new Fig. 7a).

Line 509. The thermodynamic reference level in Figure 14. Where in the figure is this indicated? Please explain. >We have explained better the thermodynamic reference level by adding: "At the onset of the thaw season which coincides with the disappearance of the snow cover, the AL exits the zero-curtain phase isothermal at 0°C (Fig. 7a). This is the thermodynamic zero level 0 MJ m−2 in Fig. 14. The sensible heat content Hθ al is zero at the onset and end of the thaw season."

Line 509. The symbol for sensible heat in this sentence is slightly different from the one appearing in the figure. Please use a consistent symbol. >Yes, we have ensured consistent notation.

Figure 14. Please spell out SEB in the figure caption, so the reader can remember what this was. >We have spelled it out.

---

## Author Response (AR2)

Innsbruck, 24.1.2025

Dear editor Jens Turowski
Dear reviewers, Benjamin Hills and anonymous reviewer #1

Our publication has greatly benefited from the thoughtful reviews and the major revision. Thank you very much for your work.

Kind regards
Dominik Amschwand on behalf of all co-authors